# CoViS-Net: A Cooperative Visual Spatial Foundation Model for Multi-Robot Applications

**Jan Blumenkamp, Steven Morad, Jennifer Gielis and Amanda Prorok**
Department of Computer Science and Technology
University of Cambridge, United Kingdom
{jb2270, sm2558, jag233, asp45}@cst.cam.ac.uk

**Abstract:** Autonomous robot operation in unstructured environments is often underpinned by spatial understanding through vision. Systems composed of *multiple* concurrently operating robots additionally require access to frequent, accurate and reliable pose estimates. In this work, we propose CoViS-Net, a decentralized visual spatial foundation model that learns spatial priors from data, enabling pose estimation as well as spatial comprehension. Our model is fully decentralized, platform-agnostic, executable in real-time using onboard compute, and does not require existing networking infrastructure. CoViS-Net provides relative pose estimates and a local bird's-eye-view (BEV) representation, even without camera overlap between robots (in contrast to classical methods). We demonstrate its use in a multi-robot formation control task across various real-world settings. We provide code, models and supplementary material online[1].

**Keywords:** Multi-Robot Systems, Robot Perception, Foundation Models

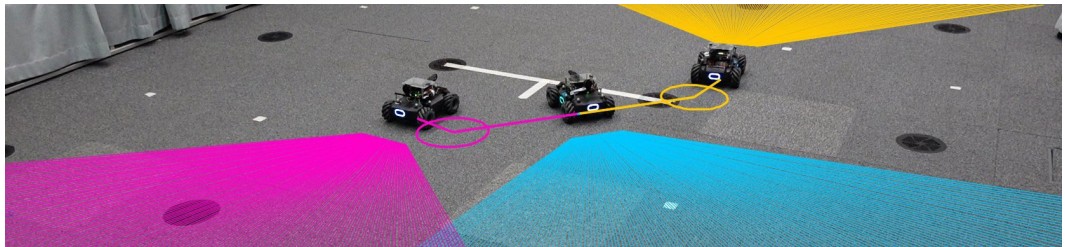

Figure 1: Our model can be used to control the relative pose of multiple follower robots (depicted in yellow and magenta) to a leader robot (blue) following a reference trajectory using visual cues in the environment only (the field-of-view is depicted as colored area). Even if there is no direct visual overlap (intersection of colored areas), the model is able to estimate the relative pose.

## 1 Introduction

Spatial understanding is a cornerstone of robotic operation in unstructured environments, relying on effective pose estimation and environmental perception [1, 2, 3, 4, 5, 6, 7, 8]. While GNSS, LiDAR, and UWB sensors have been instrumental in advancing robotics, they are limited by constraints such as indoor operation, unreliability when operating around reflective surfaces or in bright environments. *In contrast, color cameras offer a low-cost, energy-efficient, and rich data source, aligned with the vision-centric design of real-world, human-designed environments.* Classical vision-based techniques [1, 6, 5] struggle with the ill-posed nature of image processing, cannot leverage semantic priors to resolve ambiguous situations, and may converge to a solution only after multiple time steps. These challenges are exacerbated in multi-robot scenarios that require fast and accurate relative pose

---

[1]https://proroklab.github.io/CoViS-Net/

8th Conference on Robot Learning (CoRL 2024), Munich, Germany.

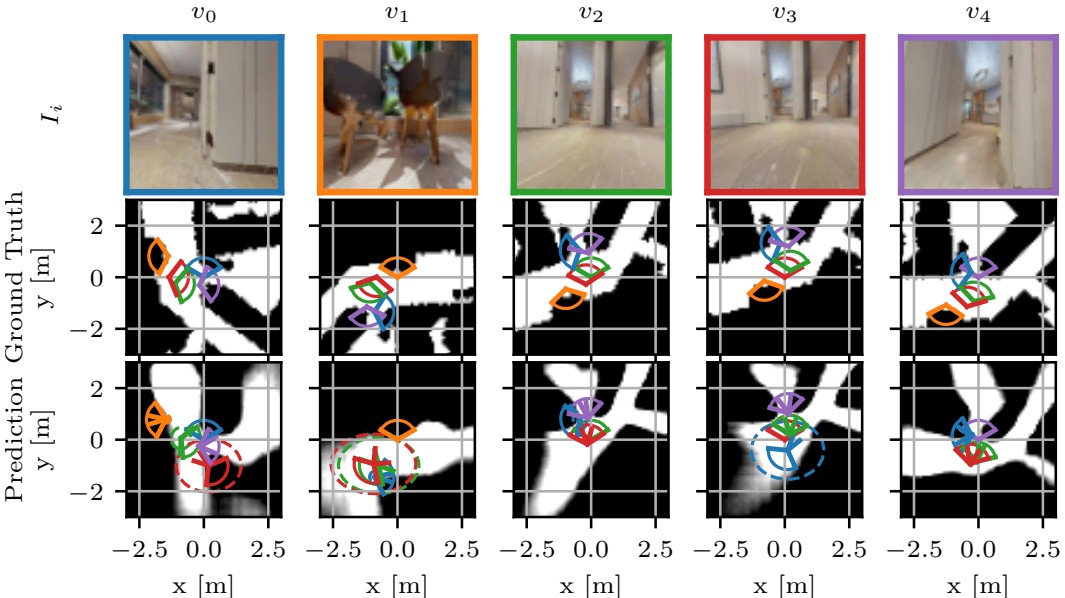

Figure 2: By leveraging spatial priors, our model provides rough relative positions *even without overlapping fields of view*, which classical approaches cannot achieve. This figure shows a sample from the simulation validation dataset $\mathcal{D}_{\text{Val}}^{\text{Sim}}$ for five nodes, each represented by a column. The top row displays the respective camera image $I_i$. The middle row shows the ground truth labels in the coordinate system $\mathcal{F}_i$, with each robot centered and facing upwards at $(0, 0)$, and the ground truth poses $\mathbf{p}_{i,ij}$ of other robots, along with their field of view. The background shows the BEV representation $\text{BEV}_i$ for each robot. The bottom row presents the corresponding predictions, displaying pose predictions $\hat{\mathbf{p}}_{i,ij}, \hat{\mathbf{R}}_{i,ij}$ with low uncertainty $\sigma_{\text{p},ij}^2 \sigma_{\text{R},ij}^2$, predicted uncertainties $\sigma_{\text{p},i}^2 ij$ as dashed oval and $\sigma_{\text{R},i}^2 ij$ similar to the FOV, and the predicted BEV representation $\hat{\text{BEV}}_i$ in the background.

estimates for tasks like flocking [9], path planning [10], and collaborative perception [11]. Explicit agent detection [4, 3, 2, 7, 8] requires line-of-sight and is not platform-agnostic, whereas deep pose predictors [12, 13, 14, 15, 16, 17, 18] show promise but have are not widely adopted in multi-robot systems due to hardware constraints. Additionally, incorporating *uncertainty metrics* is essential for assessing the reliability of predictions in these applications.

Acknowledging these limitations, we introduce CoViS-Net, a decentralized visual spatial foundation model targeting real-world multi-robot applications. *This model operates in real-time on onboard computers without the need for pre-existing network infrastructure, thus addressing critical scalability, flexibility, and robustness challenges.* CoViS-Net features a distributed architecture designed for uncertainty-aware pose estimation and BEV representation prediction in areas without local camera coverage. We demonstrate its effectiveness in real-world scenarios, including multi-robot control tasks in real-world indoor and outdoor environments.

In summary, our contributions are threefold: (1) We introduce a novel architecture for decentralized, real-time multi-robot pose estimation from monocular images, with built-in uncertainty awareness. (2) We extend it to predict BEV representations, facilitating spatial understanding in occluded areas through communication and by leveraging spatial priors. (3) We validate our model in the real-world for a multi-robot control task, demonstrating robustness even in scenarios with no image overlap.

## 2   Related Work

**Multi-robot systems** require coordination of fast-moving robots in close proximity, necessitating spatial understanding and accurate pose estimates. Key applications include flocking [9], formation control [19], trajectory deconfliction [20], multi-agent object manipulation [21, 22], area cover-

age [23, 24], and exploration [25, 26, 27, 28]. Recent advances use Graph Neural Networks (GNNs) for enhanced control and perception, enabling inter-robot communication through latent messages to address the challenge of partial observability [29, 30, 31, 32, 33, 34, 35]. Environment inference deals with predicting unobserved environmental information [36, 37]. However, a significant gap remains in *acquiring* relative pose knowledge with uncertainty estimates.

**Map-free relative pose regression** [17] estimates the relative pose between two camera images without scene-specific training. Traditional methods [38] and learning-based approaches [39, 40, 41, 42] require significant visual overlap between views. Integrating scale involves depth maps or Siamese networks [12, 43, 44, 45, 14, 13, 15]. These models lack uncertainty estimates, limiting their robotics applicability, and have not been tested in real-world, real-time deployments. Our approach provides reliable relative poses with aleatoric uncertainty estimates, beyond visual overlap constraints, using unmodified pre-trained foundation models, enabling transfer learning [46].

**BEV representations** are an effective way of providing detailed spatial comprehension by merging data from a set of sensors. Learning-based approaches have recently gained popularity [47, 48, 49, 50, 51]. Multi-agent scenarios [34, 33, 52, 35, 53, 54, 25] rely on GNSS for global pose knowledge, precluding application to indoor environments, or rely on an existing SLAM stack. Our work proposes a BEV representation predictor that utilizes relative poses predicted from monocular images, instead of known static poses of a camera rig or poses derived from external sensors.

**Foundation models** are large neural networks trained on broad data using self-supervised learning, exhibiting emergence and homogenization [55]. These models, often based on the transformer architecture [56], span several types: Large Language Models (LLMs) [57, 58], vision models [59], Vision-Language Models (VLMs) [60, 61], and Vision-Language-Action Models (VLAs) [62, 63]. Recent trends include open-source development [64, 63] and significant scaling of model parameters. Deploying these models in robotics faces challenges due to memory and latency constraints, necessitating techniques like distillation [65] and quantization [66] for real-time processing.

## 3 Problem Formulation

Given a multi-robot system, our goals are: *(i)* for each robot to predict its pose and uncertainty relative to other robots as well as corresponding embeddings using non-line-of-sight visual correspondences, and *(ii)* to use these embeddings for downstream tasks like local occupancy grid prediction. Consider a multi-robot system represented by a set of nodes $\mathcal{V}$. Each node $v_i \in \mathcal{V}$ has a position $\mathbf{p}_{w,i}$ and an orientation $\mathbf{R}_{w,i}$ in the world coordinate frame $\mathcal{F}_w$. The set of edges $\mathcal{E} \subseteq \mathcal{V} \times \mathcal{V}$ defines the communication topology and thus the graph $\mathcal{G} = (\mathcal{V}, \mathcal{E})$. For each edge $(v_i, v_j) \in \mathcal{E}$, the relative position of $v_j$ in the coordinate frame of $v_i$ (denoted as $\mathcal{F}_i$) is $\mathbf{p}_{i,ij} = \mathbf{R}_{w,i}^{-1} \cdot (\mathbf{p}_{w,j} - \mathbf{p}_{w,i})$, and the relative rotation is $\mathbf{R}_{i,ij} = \mathbf{R}_{w,i}^{-1} \cdot \mathbf{R}_{w,j}$. Each node $v_i$ is equipped with a camera $C_i$, providing an image $I_i$. From the images, we estimate relative poses and uncertainties $(\hat{\mathbf{p}}_{i,ij}, \sigma_{\mathrm{p},ij}^2, \hat{\mathbf{R}}_{i,ij}, \sigma_{\mathrm{R},ij}^2) \, \forall \, v_j \in \mathcal{N}(v_i)$ with respect to $v_i$ in its coordinate frame $\mathcal{F}_i$. This approach does not assume overlap between images or line-of-sight visibility between robots, relying exclusively on the *relationship between images*, thus making it platform-agnostic. Using these estimated poses, the robots cooperatively generate a BEV representation $\hat{\mathrm{BEV}}_i$ around each node $v_i$, which can be used for various downstream multi-robot applications.

## 4 Approach

In this section, we detail our architecture and training approach. We explain dataset generation, define our model architecture, and introduce the uncertainty estimation strategy and training protocol.

### 4.1 Model

We employ four primary models; The encoder $f_{\mathrm{enc}}$ generates a common embedding of images, $f_{\mathrm{pose}}$ estimates pairwise poses, $f_{\mathrm{agg}}$ aggregates node features, and $f_{\mathrm{BEV}}$ predicts a two-dimensional BEV

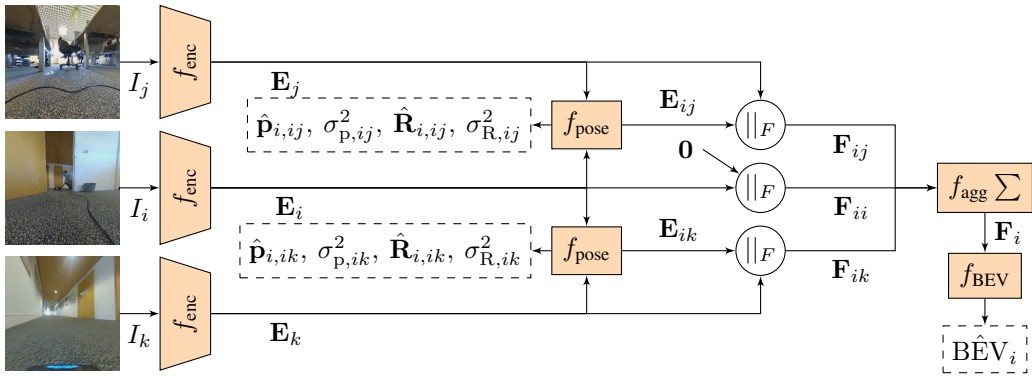

Figure 3: Overview of the model architecture: We illustrate the decentralized evaluation with respect to ego-robot $v_i$, which receives embeddings $\mathbf{E}_j, \mathbf{E}_k$ generated by the encoder $f_{\text{enc}}$ from the neighbors $v_j, v_k \in \mathcal{N}(v_i)$. For each received embedding, we employ $f_{\text{pose}}$ to compute the pose. We concatenate the pose embedding $\mathbf{E}_{ij}$ with the image embedding $\mathbf{E}_j$ (and $\mathbf{E}_{ik}$ with $\mathbf{E}_k$) and subsequently aggregate this into a node feature $\mathbf{F}_i$. Model outputs are framed with a dashed line.

representation. An overview of the model architecture is provided in Sec. 4.1. Our model uses transformers [56, 67] in $f_{\text{enc}}$, $f_{\text{pose}}$ and $f_{\text{agg}}$. We provide more details in Appendix A.1.

**Image Encoder**: We use the smallest distilled version of DinoV2 [59] (ViT-S/14) as image encoder due to its robust feature representation and spatial understanding. Each robot $v_i$ processes the image $I_i$ through $f_{\text{enc}}$ to generate the communicated node embedding $\mathbf{E}_i \in \mathbb{R}^{S \times F}$, where $S$ represents the sequence length, and $F$ the feature vector size. We freeze the weights of DinoV2, enabling the use of various open-source downstream fine-tuned image detectors without an additional forward pass.

**Pairwise pose encoder**: Each robot $v_i$) broadcasts the image encoding $\mathbf{E}_i$ to neighboring robots. Given encodings $\mathbf{E}_i$ and $\mathbf{E}_j$, $f_{\text{pose}}$ estimates edge embeddings $\mathbf{E}_{ij}$ and relative poses with uncertainties $(\hat{\mathbf{p}}_{i,ij}, \sigma^2_{\text{p},ij}, \hat{\mathbf{R}}_{i,ij}, \sigma^2_{\text{R},ij})$. For each incoming $\mathbf{E}_j \ \forall \ v_j \in \mathcal{N}(v_i)$, we concatenate the node features along the sequence dimension, add the positional embedding, run the transformer and then extract the first element of the sequence dimension to obtain the edge embedding. We then use this edge embedding to estimate position, rotation and uncertainties, resulting in $f_{\text{pose}}(\mathbf{E}_i, \mathbf{E}_j) = (\hat{\mathbf{p}}_{i,ij}, \sigma^2_{\text{p},ij}, \hat{\mathbf{R}}_{i,ij}, \sigma^2_{\text{R},ij})$.

**Multi-Node Aggregation**: We perform position-aware feature aggregation using node embeddings $\mathbf{E}_j$ from $f_{\text{pose}}$. The function $f_{\text{agg}}$ integrates information from all nodes $v_j \in \mathcal{N}(v_i) \cup v_i$, producing a feature vector $\mathbf{F}_i$ for each node $v_i$, encapsulating the aggregated information from its local vicinity. We concatenate node features along the sequence dimension and add the positional embedding. We then concatenate a nonlinear transformation of the edge features along the feature dimension. This operation is performed for all $v_j \in \mathcal{N}(v_i)$, including self-loops to handle cases where $\mathcal{N}(v_i)$ is empty by zeroing the edge embedding. We aggregate over the neighbors and the ego-robot to produce the output embedding $\mathbf{F}_i$ from all edge embeddings $\mathbf{F}_{ij}$, which is used for downstream tasks, such as estimating a BEV representation $\hat{\text{BEV}}_i = f_{\text{BEV}}(\mathbf{F}_i)$.

### 4.2 Training

For training our models, we use supervised learning. This section describes the dataset, loss functions, and uncertainty estimation.

**Dataset:** We use the Habitat simulator [68, 69] and the HM3D dataset [70], consisting of 800 scenes of 3D-scanned multi-floor buildings. The data is divided into training $\mathcal{D}^{\text{Sim}}_{\text{Train}}$ (80%), testing $\mathcal{D}^{\text{Sim}}_{\text{Test}}$ (19%), and validation $\mathcal{D}^{\text{Sim}}_{\text{Val}}$ (1%) sets. We provide more details in Appendix A.2.

**Uncertainty Estimation**: Two types of uncertainties can be modeled: *epistemic uncertainty*, which captures uncertainty in the model and can be reduced with more data, and *aleatoric uncertainty*,

which captures noise inherent to specific observations [71]. For pose estimation and pose control in robotic systems, aleatoric uncertainty is more important since downstream applications can *act* on it to *reduce* it. Traditional methods like Monte Carlo Dropout [72] and Model Ensembles [73] model epistemic uncertainty. Instead, we propose using the Gaussian Negative Log Likelihood (GNLL) Loss [74, 71], a generalization of the mean squared error (MSE) loss, allowing the model to predict both a mean $\hat{\mu}$ and a variance $\hat{\sigma}^2$, learned from data points $\mu$. The loss is defined as

$$\mathcal{L}^{\text{GNLL}}(\mu, \hat{\mu}, \hat{\sigma}^2) = \frac{1}{2} \left( \log \left( \hat{\sigma}^2 \right) + \frac{(\hat{\mu} - \mu)^2}{\hat{\sigma}^2} \right). \tag{1}$$

The GNLL loss penalizes high variance predictions and scales the MSE loss by the predicted variance, penalizing errors more when predicted uncertainty is high.

**Estimating Rotations**: Estimating rotation is difficult due to discontinuities in many standard orientation representations. Quaternions are often used for their numerical efficiency and lack of singularities. Related work [75] represents rotations with a symmetric matrix defining a Bingham distribution over unit quaternions to predict epistemic uncertainty. We combine the chordal loss and GNLL to predict aleatoric uncertainty for rotations, as shown in the following equation:

$$d_{\text{quat}}(\hat{\mathbf{q}}, \mathbf{q}) = \min \left( \|\mathbf{q} - \hat{\mathbf{q}}\|_2, \|\mathbf{q} + \hat{\mathbf{q}}\|_2 \right) \tag{2}$$

$$\mathcal{L}^2_{\text{chord}}(\hat{\mathbf{q}}, \mathbf{q}) = 2d^2_{\text{quat}}(\hat{\mathbf{q}}, \mathbf{q}) \left( 4 - d^2_{\text{quat}}(\hat{\mathbf{q}}, \mathbf{q}) \right) \tag{3}$$

$$\mathcal{L}^{\text{GNLL}}_{\text{chord}}\left( \mathbf{q}, \hat{\mathbf{q}}, \hat{\sigma}^2 \right) = \frac{1}{2} \left( \log \left( \hat{\sigma}^2 \right) + \frac{\mathcal{L}^2_{\text{chord}}(\hat{\mathbf{q}}, \mathbf{q})}{\hat{\sigma}^2} \right). \tag{4}$$

**Training**: We use a combination of the Dice loss [76] $\mathcal{L}_{\text{Dice}}$ and Binary Cross Entropy (BCE) Loss $\mathcal{L}_{\text{BEC}}$ weighted by the parameter $0 \leq \alpha \leq 1$. This is commonly referred to as *combo loss* [77] and is typically utilized for slight class imbalance. We define the total loss as

$$\mathcal{L}_{i,\text{BEV}} = \alpha \cdot \mathcal{L}_{\text{Dice}} \left( \text{BEV}_i, \hat{\text{BEV}}_i \right) + (1 - \alpha) \cdot \mathcal{L}_{\text{BCE}} \left( \text{BEV}_i, \hat{\text{BEV}}_i \right) \tag{5}$$

$$\mathcal{L}_{i,ij,\text{Pose}} = (1 - \beta) \, \mathcal{L}^{\text{GNLL}}(\mathbf{p}_{i,ij}, \hat{\mathbf{p}}_{i,ij}, \sigma^2_{\text{p},ij}) + \beta \, \mathcal{L}^{\text{GNLL}}_{\text{chord}} \left( \mathbf{R}_{i,ij}, \hat{\mathbf{R}}_{i,ij}, \sigma^2_{\text{R},ij} \right) \tag{6}$$

$$\mathcal{L} = \sum_{v_i \in \mathcal{V}} \left( \mathcal{L}_{i,\text{BEV}} + \sum_{v_j \in \mathcal{N}(v_i)} \mathcal{L}_{i,ij,\text{Pose}} \right). \tag{7}$$

The parameter $0 \leq \beta \leq 1$ balances between the position and orientation loss. Uncertainty terms are necessary for training, as many dataset samples lack overlap and may be in adjacent rooms, leading to significant errors from difficult samples that could hinder learning. We train on simulated data $\mathcal{D}^{\text{Sim}}_{\text{Train}}$ and provide further details in Appendix A.4.

## 5  Experiments and Results

We demonstrate the performance of our trained models on the test set $\mathcal{D}^{\text{Sim}}_{\text{Test}}$ and custom real-world test sets $\mathcal{D}^{\text{Real}}_{\text{Test}}$. First, we outline the metrics used. Then, we introduce the real-world dataset. Next, we analyze model performance on both simulation and real-world data. Finally, we compile and deploy our model in a real-world, real-time trajectory tracking task.

**Metrics:** We compute the Euclidean distance between the predicted and ground truth poses as $D_{\text{pos}}(\mathbf{p}_{i,ij}, \hat{\mathbf{p}}_{i,ij})$ and the geodesic distance between rotations as $D_{\text{rot}}(\mathbf{R}_{i,ij}, \hat{\mathbf{R}}_{i,ij})$ for all edges and report median errors. We categorize the median pose error into four categories. *All* includes all edges. *Visible* and *Invisible* edges are determined by FOV overlap (an edge is invisible if $D_{\text{rot}}(\mathbf{R}_{i,ij}, \mathbf{0}) > \text{FOV}$). *Invisible Filtered* samples use the predicted position uncertainty $\sigma^2_{\text{p},ij}$ to reject high-uncertainty pose estimations based on the Youden's index [78]. For simulation results, we also report the Dice and IoU metric. We provide more details in Appendix B.1.

Table 1: Ablation study over the number of patches $S$ and size of features $F$ per patch. We report the BEV representation performance and the median error for poses on the dataset $\mathcal{D}_{\text{Test}}^{\text{Sim}}$ and $\mathcal{D}_{\text{Test}}^{\text{Real}}$.

| Model | | $\mathcal{D}_{\text{Test}}^{\text{Sim}}$ | | | | $\mathcal{D}_{\text{Test}}^{\text{Real}}$ | | | | | |
|---|---|---|---|---|---|---|---|---|---|---|---|
| S | F | Dice | IoU | All | | Invis. Filt. | | Invisible | | Visible | |
| 256 | 48 | **69.1** | **57.1** | **36 cm** | 8.3° | 61 cm | **6.8°** | **97 cm** | 7.9° | 33 cm | 5.8° |
| 128 | 96 | 68.8 | 56.8 | 40 cm | 8.4° | **55 cm** | 7.7° | 97 cm | **7.4°** | 32 cm | **5.6°** |
| 128 | 48 | 67.9 | 56.1 | 38 cm | **7.7°** | 67 cm | 9.9° | 113 cm | 9.6° | **29 cm** | 5.7° |
| 128 | 24 | 66.7 | 54.6 | 50 cm | 9.5° | 83 cm | 11.2° | 112 cm | 9.7° | 31 cm | 5.7° |
| 64 | 48 | 61.0 | 43.5 | 51 cm | 9.6° | 81 cm | 9.4° | 119 cm | 10.8° | 36 cm | 6.3° |
| 1 | 3072 | 47.0 | 1.4 | 144 cm | 89.9° | 123 cm | 25.7° | 122 cm | 25.8° | 93 cm | 138.2° |
| 1 | 348 | 47.1 | 1.4 | 84 cm | 11.7° | 150 cm | 164.1° | 135 cm | 37.9° | 80 cm | 11.1° |

## 5.1 Dataset Evaluation

We conducted experiments across various model variants trained with different parameters, reporting on our model's efficiency. We present quantitative results in Tab. 1 and qualitative findings in Fig. 2.

**Real-World Dataset:** Our setup includes four Cambridge RoboMasters [79] and one Unitree Go1 Quadruped, each equipped with local compute, a monocular camera and WiFi dongles for adhoc networking. Our custom real-world test dataset $\mathcal{D}_{\text{Test}}^{\text{Real}}$ covers a diverse and challenging range of indoor and outdoor scenes. We provide more details in Appendix A.3.

**Results:** Tab. 1 details the performance across seven different experiments. We evaluated the impact of image embedding sizes $\mathbf{E}_i$ on the performance. The IoU and Dice metrics for the BEV representation, alongside median pose estimates for all samples in the dataset, are provided. Experiments show that the sequence length $S$ has a more significant influence on BEV representation and pose estimate performance than the feature size $F$. We set $S$ to 1 for two experiments: *(1)* increasing $F$ to match the size of the embeddings for other experiments, and *(2)* decreasing $F$, effectively reducing the transformer to a CNN-only architecture by removing the attention mechanism. Both experiments result in reduced performance, highlighting the importance of attention in our model. The best-performing model with $S = 256$ achieves an IoU of $0.571$, a Dice score of $0.691$, and median pose accuracy of $36$ cm and $8.3°$, which is roughly equal to one robot body length. In contrast, models with $S = 1$ show significant degradation, with nearly unusable BEV representation predictions and pose errors on the scale of meters. Models with $F = 24$ and $S = 64$ performed slightly worse than those with larger $F$ or $S$. We provide additional results in Appendix B.2. Evaluation on the real-world dataset $\mathcal{D}_{\text{Test}}^{\text{Real}}$ focuses on position and rotation accuracy, excluding quantitative BEV performance due to unavailable ground truth data. The results align with $\mathcal{D}_{\text{Test}}^{\text{Sim}}$, showing a correlation between the number of patches $S$ and features $F$ and pose estimation performance. The rotation estimates do not significantly differ between visible and invisible samples, demonstrating robust pose estimation without requiring visual overlap. The most accurate model achieves a median localization error of $97$ cm and $7.9°$ for invisible samples, and $33$ cm and $5.8°$ for visible samples. Note that $\mathcal{D}_{\text{Test}}^{\text{Real}}$ samples are at most 2 m apart, resulting in a worst-case error of up to 4 m. These findings demonstrate our model's capability to reliably estimate poses, and accurately predict higher uncertainties under adverse conditions. We attribute this performance to our pose prediction model, as well as the robust spatial representation capabilities of the DinoV2 encoder. The model goes beyond mere feature matching, achieving sophisticated scene understanding that allows it to "imagine" unseen portions of the scene based on learned data priors, thus facilitating pose estimation through *environment inference* [36, 37]. We provide additional results in Appendix B.3.

Finally, we conduct a baseline comparison with two feature-matching-based approaches, as summarized in Table 2. Both baseline methods extract and match features across two images to estimate the essential matrix, deriving rotation and scale-less translation vectors. The first baseline uses OpenCV for ORB feature extraction followed by brute-force matching. The second baseline, LightGlue, uses a neural network-based approach for feature extraction and matching. Given both baselines estimate scale-less transformations, we adopt the same metric used in LightGlue [42], computing the AUC

Table 2: We report the FPS and AUC metric at 20, 45 and 90°on two baselines.

| AUC@ | FPS | All | | | Invisible | | | Visible | | |
|---|---|---|---|---|---|---|---|---|---|---|
| | | 20 | 45 | 90 | 20 | 45 | 90 | 20 | 45 | 90 |
| ORB/OpenCV [80] | 2.79 | 6.45 | 15.43 | 28.26 | 0.16 | 1.60 | 7.43 | 9.41 | 21.92 | 38.03 |
| LightGlue [42] | 1.17 | 16.30 | 27.14 | 37.63 | 0.02 | 0.09 | 0.31 | 23.94 | 39.82 | 55.14 |
| Ours | **42.73** | **25.23** | **47.63** | **66.34** | **9.22** | **24.40** | **45.74** | **32.74** | **58.53** | **76.01** |

at thresholds of 20, 45, and 90 degrees based on the maximum error in rotation and translation. Our findings indicate that our model surpasses both baselines in performance and speed across image pairs with and without visual overlap. Notably, while both baselines require visual overlap to function effectively, our approach does not, highlighting its robustness and applicability in a broader range of scenarios.

## 5.2 Real-World Pose Control

In this section, we describe our experiment setup (including model deployment, wireless communication, and controllers), demonstrating real-world multi-robot control tasks.

**Model deployment**: We deploy a model with $F = 24$ and $S = 128$. After training in PyTorch, we compile $f_{enc}$, $f_{pose}$, and $f_{agg}$ for decentralized deployment. Using TensorRT, we achieve sub-30 ms processing times. We provide more details and a runtime analysis in Appendix B.4.

**Ad-hoc WiFi**: Image embeddings are sent between robots at 15 Hz, each over 6 KiB. The system's distributed nature suits broadcast communication, allowing scalable operation. However, IEEE 802.11/WiFi's fallback to low bit rates for broadcast increases frame loss probability, worsening with more robots. Higher-level systems like ROS2's CycloneDDS manage retransmissions but can't detect network overloads, leading to increased packet loss. We design a custom protocol that dynamically adjusts messaging load based on network conditions and uses TDMA scheduling to maximize frame reception without compromising data rates. We provide more details in Appendix B.4.

**Experiment**: We conduct qualitative experiments in various environments, including outdoors, and quantitative experiments. Two follower robots maintain a fixed orientation and distance from a leader robot on a predefined trajectory. A PD controller manages relative pose control, adjusting based on predicted uncertainty and selectively deactivating if uncertainty thresholds are exceeded. This ensures that even with unreliable estimates, robots maintain orientation toward the leader until reliable estimates are available. We assess tracking accuracy by comparing ground truth and estimated poses, focusing on position and rotation errors and predicted uncertainties.

**Results**: Fig. 5 shows snapshots from four different real-world deployments, and Fig. 4 shows the trajectories of all robots for all quantitative experiments, as well as a visualization of the position and rotation error over time. We provide additional results in Appendix B.4.

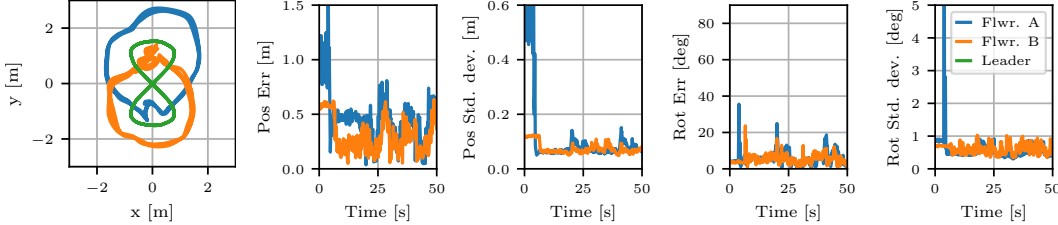

Figure 4: We evaluate the tracking performance of our model and uncertainty-aware controller on reference trajectories with two follower robots (blue and orange) positioned to the left and right of the leader robot (green). We report tracking performance over time for position and rotation, as well as the predicted uncertainties. The leader always faces the direction of movement. We show the trajectory for 120 s and the tracking error for the first 60 s.

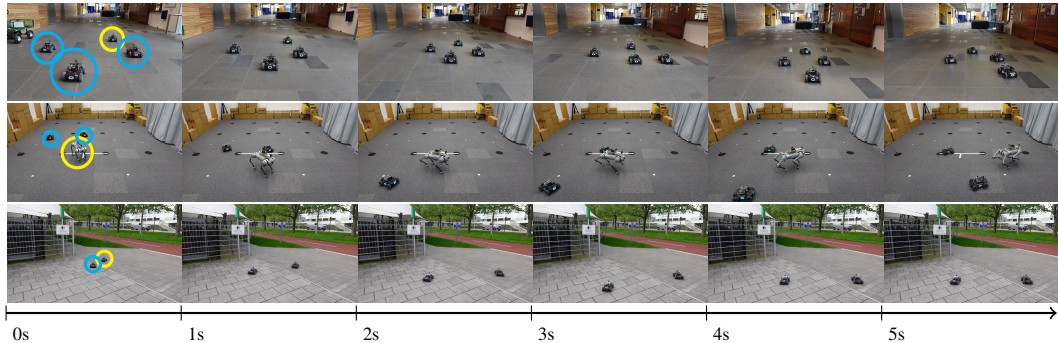

0s    1s    2s    3s    4s    5s

Figure 5: We show snapshots of real-world deployments in different scenes with up to four robots. Each row contains six frames from a video recording, each spaced 1 s in time. We indicate the positions of the robots in the first frame, where the leader is circled yellow and the followers blue. The first three samples show indoor scenes, and the bottom sample is an outdoor deployment. The second sample shows a heterogeneous deployment, combining robots of different sizes and dynamics.

The qualitative experiments show a robust system that generalizes to a wide range of scenarios, including outdoors, albeit with a scale prediction that is less reliable outdoors. Our model was trained exclusively on indoor data and generalizes to outdoor data due to the wide range of training data used on the DinoV2 encoder. We furthermore show platform agnosticism by deploying our model to a heterogeneous team of robots, lead by a quadruped (second row in Fig. 5).

In quantitative experiments, the tracking error oscillates throughout the trajectories as the followers reactively adjust their positions to the leader's movements. Follower A starts facing the opposite direction of the leader, resulting in high position variance and low rotation variance, which decreases as the robot aligns with the leader. An erroneous rotation prediction is accompanied by high uncertainty. As the leader moves, one follower is inside and the other outside the trajectory, switching roles depending on the section of the figure. We report a median tracking error of 48 cm and 5.3° for Follower A, and 26 cm and 5.4° for Follower B (corresponding roughly to one robot body length).

## 6   Limitations

Foundation models are commonly trained *unsupervised* [55], whereas our model is trained on the HM3D indoor dataset in a *supervised* manner. Our model requires peer-to-peer communication to be available to merge information from other robots and GPU-accelerated onboard computing for real-time deployment. Previous work showed the scalability of GNNs due to the principle of locality (number of neighboring nodes are constrained by the communication range) [29]. The number of real robots is currently constrained by our custom networking stack.

## 7   Conclusion

We introduced a first-of-its-kind cooperative visual spatial foundation model that leverages existing large vision models and demonstrated its use in multi-robot control. From monocular camera images only, our model is able to output accurate relative pose estimates and BEV representations by aggregating information across neighboring robots. We validated our model on a custom real-world test set and demonstrated real-time multi-robot control. Our model accurately predicts poses, even without pixel-level correspondences, and correctly predicts BEV representations for obscured regions. In further real-world tests, we demonstrated accurate trajectory tracking, paving the way for more complex vision-only robot tasks. Our model estimates relative poses with a median error of 33 cm and 5.8° for visible edges, and 97 cm and 5.9° for invisible edges, roughly equivalent to one body length of our robot. In future work, we plan to include temporal data, apply this model to other downstream tasks and combine it with learning-based multi-agent control policies.

## Acknowledgments

This work was supported in part by European Research Council (ERC) Project 949949 (gAIa) and in part by ARL DCIST CRA W911NF-17-2-0181. J. Blumenkamp acknowledges the support of the 'Studienstiftung des deutschen Volkes' and an EPSRC tuition fee grant. We also acknowledge a gift from Arm. We gratefully acknowledge Toshiba Europe Ltd, Cambridge Research Laboratory for providing the Unitree Go1 used in our real-world experiments.

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

# Appendices

## A  Model Training

This section details our model architecture, datasets, and training process. We describe the key components of our model, explain the creation of our training and validation datasets, and outline our training methodology.

### A.1  Model

This section introduces all hyperparameters and implementation details of the model architecture.

Our methodology utilizes the Transformer architecture [56]. Self-attention computes the output as a weighted sum of the values ($\mathbf{V}$), with the weights determined by the compatibility of queries ($\mathbf{Q}$) and keys ($\mathbf{K}$) as per the formula: $\text{Attention}(\mathbf{Q}, \mathbf{K}, \mathbf{V}) = \text{softmax}((\mathbf{Q}\mathbf{K}^T)/\sqrt{d_k})\mathbf{V}$, where $d_k$ is the dimensionality of the keys. Typically, $\mathbf{Q}$, $\mathbf{K}$, and $\mathbf{V}$ are set to the same input features.

DinoV2 is a vision foundation model based on the vision transformer (ViT) and pre-trained in a self-supervised manner on the LVD-142M dataset, which is a cumulation of various filtered datasets, including various versions of ImageNet [81] and Google Landmarks.

The encoder $f_\text{enc}$ consists of the pre-trained DinoV2 vision transformer and two more transformer layers previously described as $f_\text{trans}^\text{enc}$. We use the smallest distilled version of DinoV2 based on ViT-S/14 with 22.1 M parameters due to the fastest inference on our hardware. The input feature size is $224 \times 224$, of which DinoV2 generates an embedding of size $S = 256$ and $F = 384$. We add a linear layer that transforms $F$ down to the parameter described in this paper (e.g. 24) while maintaining $S$. The output of this is then cropped to the $S$ described in this paper (e.g. 128) by cropping the sequence length at that position.

$$f_\text{enc} = f_\text{dino} \cdot f_\text{trans}^\text{enc} \tag{8}$$

$$f_\text{enc}(I_i) = \mathbf{E}_i \tag{9}$$

The pairwise pose encoder $f_\text{pose}$ consists of transformer $f_\text{trans}^\text{pose}$, which is assembled from five transformer blocks, and four MLPs $f_\text{p}^\mu$, $f_\text{p}^\sigma$, $f_\text{R}^\mu$, $f_\text{R}^\sigma$.

We define the concatenation operation of two tensors $\mathbf{A}$ and $\mathbf{B}$ along axis $C$ as $(\mathbf{A} \,||_C\, \mathbf{B})$ and the indexing operation on tensor $\mathbf{A}$ by the first element along axis $B$ as $\mathbf{A}_{B1}$. We introduce an instance of another transformer $f_\text{trans}^\text{pose}$, and four MLPs $f_\text{p}^\mu$, $f_\text{p}^\sigma$, $f_\text{R}^\mu$, $f_\text{R}^\sigma$. Furthermore, we introduce a learnable transformer positional embedding for the sequence dimension $\mathbf{S}_\text{pose}$. For each incoming $\mathbf{E}_j \,\forall\, v_j \in \mathcal{N}(v_i)$, we perform a pose estimation as

$$\mathbf{E}_{ij} = f_\text{trans}^\text{pose}\left((\mathbf{E}_i \,||_S\, \mathbf{E}_j) + \mathbf{S}_\text{pose}\right)_{S1} \tag{10}$$

$$\hat{\mathbf{p}}_{i,ij},\ \hat{\mathbf{R}}_{i,ij} = f_\text{p}^\mu(\mathbf{E}_{ij}),\ f_\text{R}^\mu(\mathbf{E}_{ij}) \tag{11}$$

$$\sigma_{\text{p},ij}^2,\ \sigma_{\text{R},ij}^2 = f_\text{p}^\sigma(\mathbf{E}_{ij}),\ f_\text{R}^\sigma(\mathbf{E}_{ij}) \tag{12}$$

$$f_\text{pose}(\mathbf{E}_i, \mathbf{E}_j) = (\hat{\mathbf{p}}_{i,ij}, \hat{\mathbf{R}}_{i,ij}, \sigma_{\text{p},ij}^2, \sigma_{\text{R},ij}^2).$$

We concatenate the node features along the sequence dimension, add the positional embedding, run the transformer and then extract the first element of the sequence dimension (Eq. 10), and eventually use this edge embedding to estimate poses and orientations (Eq. 11) and uncertainties (Eq. 12). We first project the embedding $\mathbf{E}_i$ to a larger feature size $F = 192$ using a linear layer. Each transformer layer has $F = 192$ in and out features. The learnable transformer positional embedding $\mathbf{S}_\text{pose}$ is initialized to match $F$ and $S$, with normal distribution initialization of $\sigma = 0.02$. The MLPs $f_\text{p}^\mu$, $f_\text{p}^\sigma$, $f_\text{R}^\mu$, $f_\text{R}^\sigma$ are represented by one linear layer mapping the output of the transformer from $F = 192$ to $F = 17$ (position: 3, position uncertainty: 3, orientation: 10, orientation uncertainty: 1).

**Multi-node aggregation**: Lastly, we perform the position-aware feature aggregation, utilizing the node embeddings $\mathbf{E}_j$ generated in $f_\text{pose}$ for each corresponding edge. The feature aggregation function $f_\text{agg}$ integrates information from the set of nodes $v_j \in \mathcal{N}(v_i) \cup \{v_i\}$. The resulting output is a

feature vector $\mathbf{F}_i$ for each node $v_i$, encapsulating the aggregated information from its local vicinity, which can be used for any downstream task in a transfer-learning context [46], requiring geometric information in a robotic swarm, for example, to predict a BEV of the environment around the robot that captures areas that might not necessarily be visible by the robot itself.

We introduce an instance of two more transformers $f_{\text{trans}}^{\text{agg}}$ and $f_{\text{trans}}^{\text{post}}$, an MLP $f_{\text{aggpos}}$, a learnable transformer positional embedding $\mathbf{S}_{\text{agg}}$, as well as a BEV decoder $f_{\text{BEV}}$. We perform the node aggregation as

The feature aggregation function $f_{\text{agg}}$ is built from $f_{\text{trans}}^{\text{agg}}$, which consists of five transformer layers, and $f_{\text{trans}}^{\text{post}}$, which consists of one transformer layer, all of which have an embedding size of $F = 192$. The MLP $f_{\text{aggpos}}$ consists of three layers with 17, 48 and 48 neurons each, the positional embedding $\mathbf{S}_{\text{agg}}$ is initialized similar to the one in $f_{\text{pose}}$, and the BEV decoder $f_{\text{BEV}}$ takes the first sequence element of $f_{\text{trans}}^{\text{post}}$ as input and scales it up through seven alternations of convolutions and upsampling.

$$\mathbf{X}_{ij} = ((\mathbf{E}_i \,||_S\, \mathbf{E}_j) + \mathbf{S}_{\text{agg}}) \tag{13}$$

$$\mathbf{F}_{ij} = (\mathbf{X}_{ij} \,||_F\, f_{\text{aggpos}}(\mathbf{E}_{ij})) \tag{14}$$

$$\mathbf{F}_{ii} = (\mathbf{X}_{ii} \,||_F\, f_{\text{aggpos}}(\mathbf{0})) \tag{15}$$

$$\mathbf{F}_i = f_{\text{trans}}^{\text{post}} \left( \sum_{v_j \in \mathcal{N}(v_i) \cup \{v_i\}} f_{\text{trans}}^{\text{aggr}}(\mathbf{F}_{ij}) \right)_{S1} \tag{16}$$

$$\hat{\text{BEV}}_i = f_{\text{BEV}}(\mathbf{F}_i). \tag{17}$$

We concatenate the node features along the sequence dimension and add the positional embedding (Eq. 13). We then concatenate a nonlinear transformation of the edge features along the feature dimension to the output of the previous operation (Eq. 14). The previous operations are performed for all $v_j \in \mathcal{N}(v_i)$, but it is essential to include self-loops to the graph topology, as a full forward pass of the neural network would otherwise not be possible if $\mathcal{N}(v_i)$ were empty. We therefore set the edge embedding to the zero tensor $\mathbf{0}$ in that case (Eq. 14). We then aggregate over the set of neighbors and itself to produce the output embedding (Eq. 16) and finally use this to estimate any other downstream task, for example, estimating a BEV (Eq. 17).

All transformer blocks have 12 attention heads. The MLP used in the transformer has a hidden size of $4F$. All blocks utilize a dropout of $0.2$. The model has a total of $43.7$ M parameters ($22.1$ M of which are frozen DinoV2 parameters and $21.6$ M trainable parameters).

## A.2 Training Dataset

We utilize the Habitat simulator [68, 69] along with the training set of the HM3D dataset [70], which comprises 800 scenes of 3D-scanned real-world multi-floor buildings. We select Habitat and HM3D for their photorealism and the availability of ground-truth pose information, as well as detailed navigation mesh information that can be converted to a BEV representation of the scene. We instantiate a single agent equipped with a calibrated, rectified camera with a field of view of $120°$, which corresponds to the camera configuration on our robots. For each scene, we randomly sample several uniform agent positions based on the total area of the scene. For each sample, the camera orientation is randomized across all three axes. Additionally, for each scene, we extract the processed navigation mesh, which indicates areas that are navigable by a mobile robot, or not navigable due to obstacles or overhangs. We extract $3,816,288$ images from a calibrated camera with a $120°$ field of view from 800 scenes and $1,411$ floors, along with corresponding BEV representations.

The samples of the datasets $\mathcal{D}_{\text{Train}}^{\text{Sim}}$ $\mathcal{D}_{\text{Test}}^{\text{Sim}}$ and $\mathcal{D}_{\text{Val}}^{\text{Sim}}$ are randomly sampled to be within orientation range $[-\frac{\pi}{32}, \frac{\pi}{32}]$ for roll and pitch, and within a height of $0.0$ to $2.0$ meters, or less if the ceiling is at a lower height.

The samples of the datasets $\mathcal{D}_{\text{Train6D}}^{\text{Sim}}$ $\mathcal{D}_{\text{Test6D}}^{\text{Sim}}$ are randomly sampled to be within orientation range $[-\pi, \pi]$ for roll and pitch. We furthermore randomize the FOV ranging from $60°$ to $120°$ per camera.

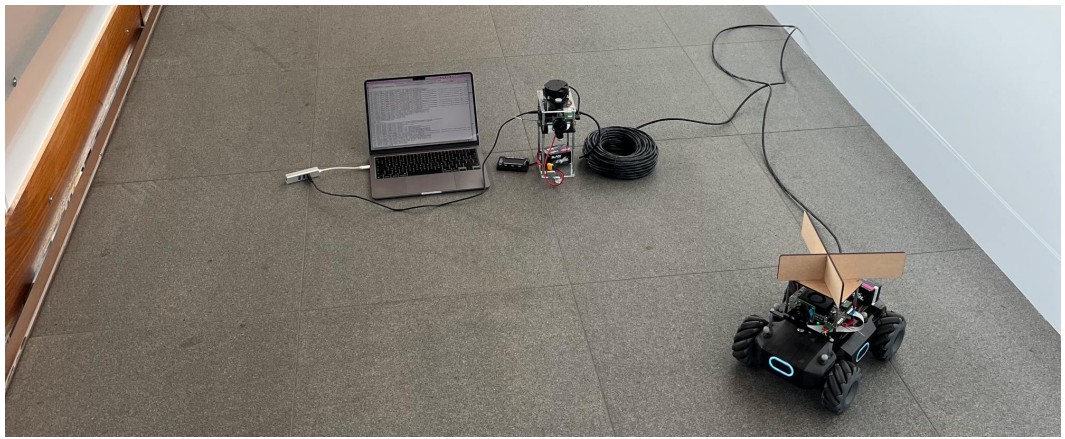

Figure 6: To collect the real-world testset, we construct a static base station equipped with two cameras facing in opposite directions and a lidar. We add a marker on the robot so that its pose can be tracked with the lidar and then manually move the robot around the base station.

We divide our data into training $\mathcal{D}_{\text{Train}}^{\text{Sim}}$ (80%), testing $\mathcal{D}_{\text{Test}}^{\text{Sim}}$ (19%), and validation $\mathcal{D}_{\text{Val}}^{\text{Sim}}$ (1%) sets. From these sets, we sample pairs of $N_{\max} = 5$ images and poses by (1) uniformly sampling 25% of individual images from each dataset, and then (2) uniformly sampling $N_{\max} - 1$ images within a distance of $d_{\max} = 2$ m around each sample from (1). This process yields $763,257$ training and $276,680$ test tuples, each of which are at most $2d_{\max}$ apart and pointing in random directions, thus providing realistically close and distant data samples for estimating both poses and uncertainties in case of no overlaps.

The ground truth BEV $\text{BEV}_i$ is a cropped 6 m × 6 m view of the floor's BEV around $v_i$, generated through Habitat's navigation mesh. Each agent $v_i$ is centered and facing north.

### A.3 Validation Dataset

Data collection involves a base station with dual cameras and a 2D Lidar, plus a distinctively shaped marker on each robot. We manually navigated robots around the base station to capture a variety of relative poses, then used a custom tool to label Lidar data for pose extraction.

All robots are equipped with an NVidia Jetson Orin NX 16GB and a forward-facing camera, rectified to a 120° FOV with an image resolution of $224 \times 224$ pixels. We collected the real-world test set $\mathcal{D}_{\text{Test}}^{\text{Real}}$ in eight indoor scenes. It consists of 5692 images. We sample image pairs to ensure a uniform distribution of relative positions from 0 m to 2 m. This results in 14008 sets of samples, each with three images, totaling 84048 pose edges, 32% which have no visual overlap. Our dataset covers a diverse and challenging range of indoor scenes, as well as one outdoor scene.

We explain the setup of our data collection unit in Fig. 6. We visualize the five different scenes we collected the dataset from as well as the data distribution from each respective scene in Fig. 7.

### A.4 Training

We use Pytorch Lightning to train our model for 15 epochs on two NVidia A100 GPUs. This training takes approximately 24 hours. We optimize the weights using the AdamW optimizer and use the 1cycle learning rate annealing schedule throughout the training and configured weight decay. We choose the best model based on the performance on the validation set.

We set the initial learning rate to $1^{-5}$ and configure a learning rate schedule to increase this over two epochs up to $1^{-3}$, followed by a sinusoidal decline to 0 over 20 epochs.

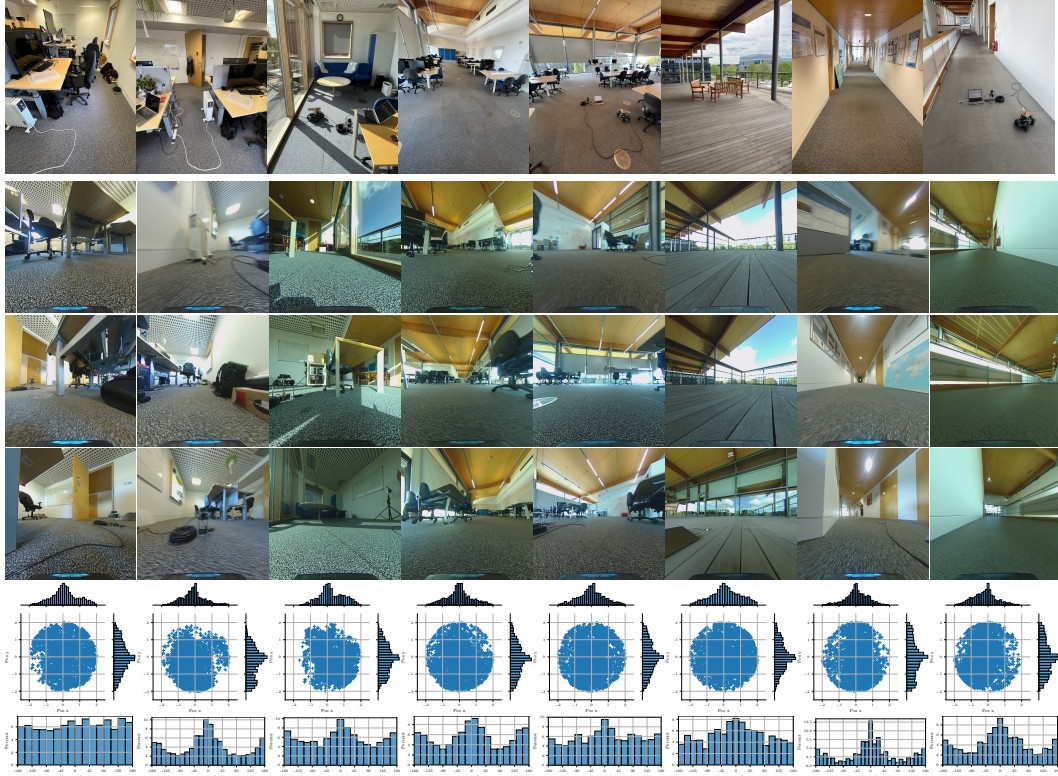

Figure 7: We collect the real-world dataset $\mathcal{D}_{\text{Test}}^{\text{Real}}$ from eight unique scenes, showing different challenges, from cluttered environments over scenes with high ceilings to sunny floors. The first row shows a picture summary of the scene, taken from first-person-view perspective. The second to fourth row shows three samples from the real-world dataset, from the perspective of the robot. The fifth row shows the distribution of positions for each scene and the sixth row shows the distribution of relative angles between nodes.

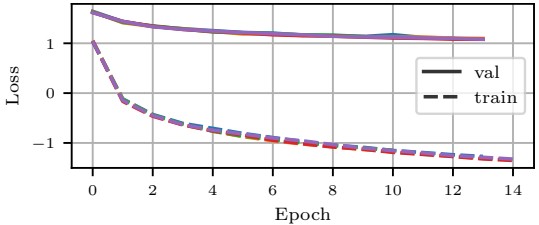

Figure 8: We show the training and validation loss for a model trained with $F = 48$ and $S = 128$ over $N = 5$ different training seeds. We report a low average standard deviation of 0.011374 over all epochs on the validation loss.

We show the train and validation loss for $N = 5$ training runs with different seeds in Fig. 8. Training with $F = 48$ and $S = 128$ across $N = 5$ different seeds reveals a low average standard deviation of 0.011374, leading us to report results for only $N = 1$ seeds in Tab. 1.

The parameters $\alpha$ balances between Dice and BCE loss, and we set it to $\alpha = 0.5$, and the parameter $\beta$ balances between position and orientation prediction accuracy, and we set it to 1, given equal weight to both.

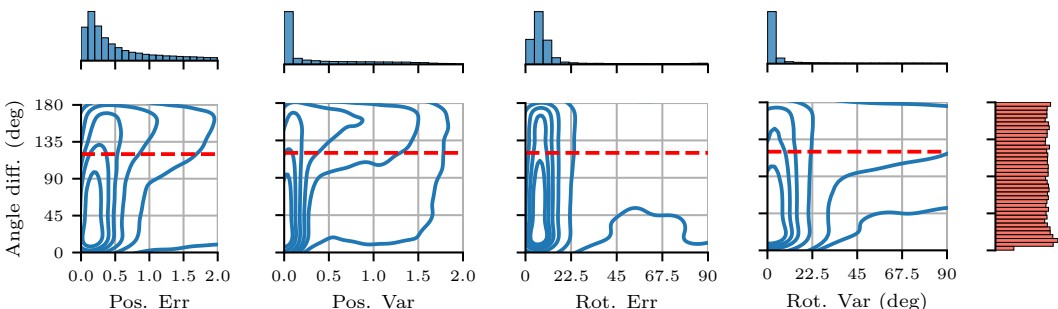

Figure 9: We show qualitative distributions for position and rotation errors and variances over the relative angle between two robots on all edges of the simulation testset $\mathcal{D}_{\text{Test}}^{\text{Sim}}$. Two agents facing the same direction would have an angle difference of $0°$, whereas two agents facing the opposite direction would have an angle difference of $180°$. We show the FOV as dashed line, where any results below the line indicate that the edge has some image overlap and above no image overlap. The horizontal marginals show the distribution of each individual plot while the vertical marginal shows the distribution of angle differences over the dataset. While the position error and variance increase over the threshold, the majority of the predictions are still useful. The rotation error is consistent across angle differences, while the uncertainty increases.

Table 3: Ablation over different modes for the BEV prediction on the simulation testset $\mathcal{D}_{\text{Test}}^{\text{Sim}}$.

| Experiment | Dice | IoU | Median Pose Err. |
|---|---|---|---|
| None | 0.628 | 0.495 | N/A |
| Predicted | 0.683 | 0.561 | 31 cm, 5.0° |
| Ground truth | 0.743 | 0.632 | 0 cm, 0.0° |

# B  Experiments and Results

## B.1  Metrics

We compute the absolute Euclidean distance between the predicted and ground truth poses and the geodesic distance between rotations as

$$D_{\text{pos}}(\mathbf{p}_{i,ij}, \hat{\mathbf{p}}_{i,ij}) = \|\mathbf{p}_{i,ij} - \hat{\mathbf{p}}_{i,ij}\|$$

$$D_{\text{rot}}(\mathbf{R}_{i,ij}, \hat{\mathbf{R}}_{i,ij}) = 4 \cdot \arcsin\left(\frac{1}{2} d_{\text{quat}}(\hat{\mathbf{R}}_{i,ij}, \mathbf{R}_{i,ij})\right)$$

for all edges and report median errors.

We perform a further evaluation on the distributions of pose errors and variations on the simulation test set $\mathcal{D}_{\text{Test}}^{\text{Sim}}$ in Fig. 9. The results are in line with the results performed on $\mathcal{D}_{\text{Test}}^{\text{Real}}$ in the main manuscript.

## B.2  Simulation

We provide additional quantitative experiments in Fig. 9 and Tab. 3. We visualize four additional qualitative samples from the simulation dataset $\mathcal{D}_{\text{Test}}^{\text{Sim}}$ in Fig. 10, Fig. 11, Fig. 12, Fig. 13, Fig. 14 and Fig. 15.

**BEV Experiments:** We assess the effectiveness of BEV predictions, focusing on the contribution of pose predictions to enhancing BEV accuracy, in an ablation study detailed in Tab. 3.

We conduct three experiments: *(1)* A model trained without incorporating any pose prediction into the BEV aggregator. This model can estimate local information and, to a degree, information aggregated through neighbors. However, it must implicitly learn to predict the relative poses between observations to successfully merge the BEV representations from multiple agents. *(2)* A model

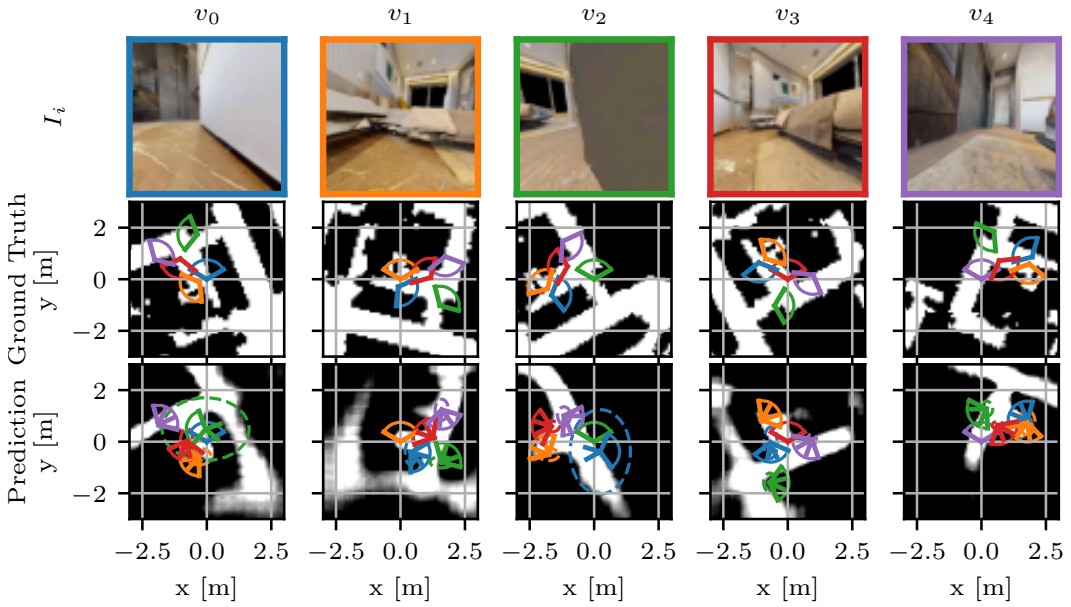

Figure 10: Sample A from $\mathcal{D}_{\text{Test}}^{\text{Sim}}$.

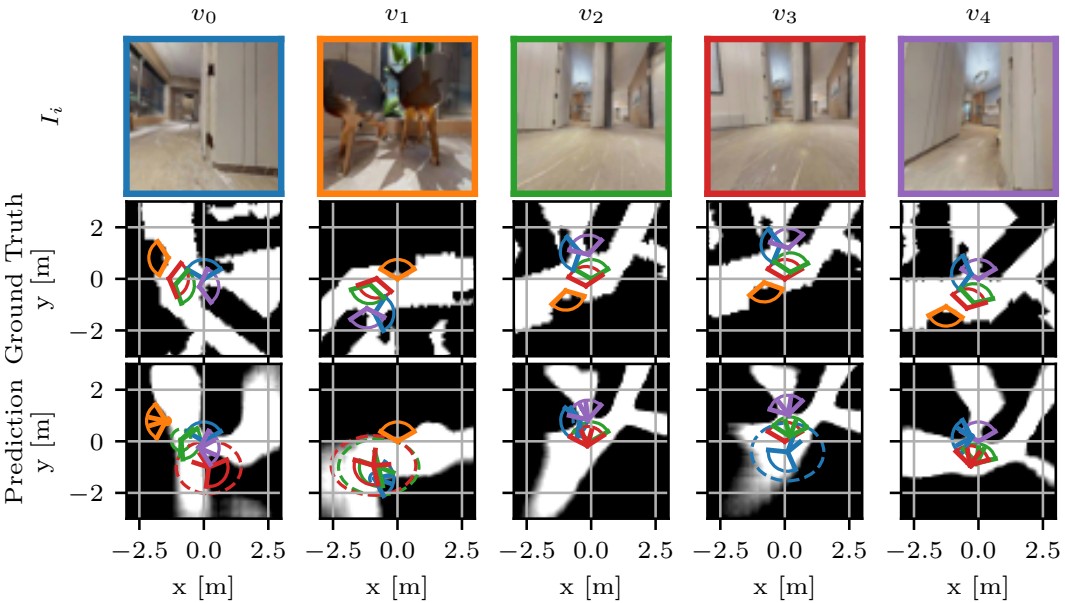

Figure 11: Sample B from $\mathcal{D}_{\text{Test}}^{\text{Sim}}$.

trained as described in this work, integrating pose estimates with BEV predictions and aggregating this information from neighbor nodes. For this experiment, we use the model configuration with $F = 48$ and $S = 128$. *(3)* A BEV model trained with ground truth pose information from the simulation training set $\mathcal{D}_{\text{Train}}^{\text{Sim}}$. This model's performance is considered an upper limit, assuming perfect pose estimation. Evaluation on the simulation test set $\mathcal{D}_{\text{Test}}^{\text{Sim}}$ indicates our model performs between the two baselines. It significantly outperforms the pose-free model with an $8.75\%$ improvement in accuracy, while the model using ground truth poses shows an $18.31\%$ performance increase compared to the baseline without pose information. These results highlight the substantial benefit of incorporating pose predictions into BEV estimation, positioning our approach as a notably effective middle ground between the two extremes.

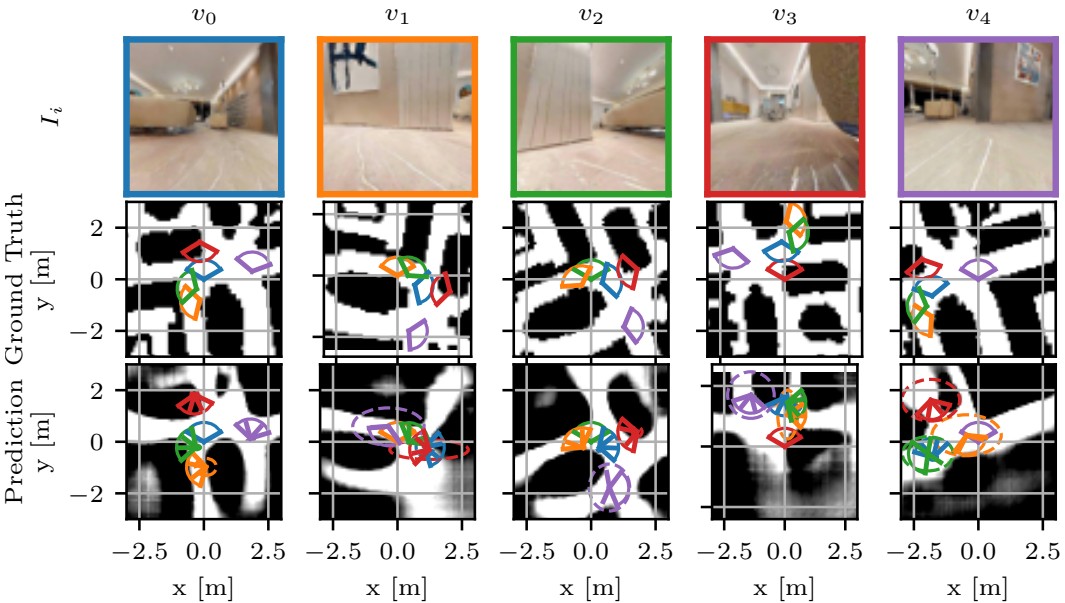

Figure 12: Sample C from $\mathcal{D}_{\text{Test}}^{\text{Sim}}$.

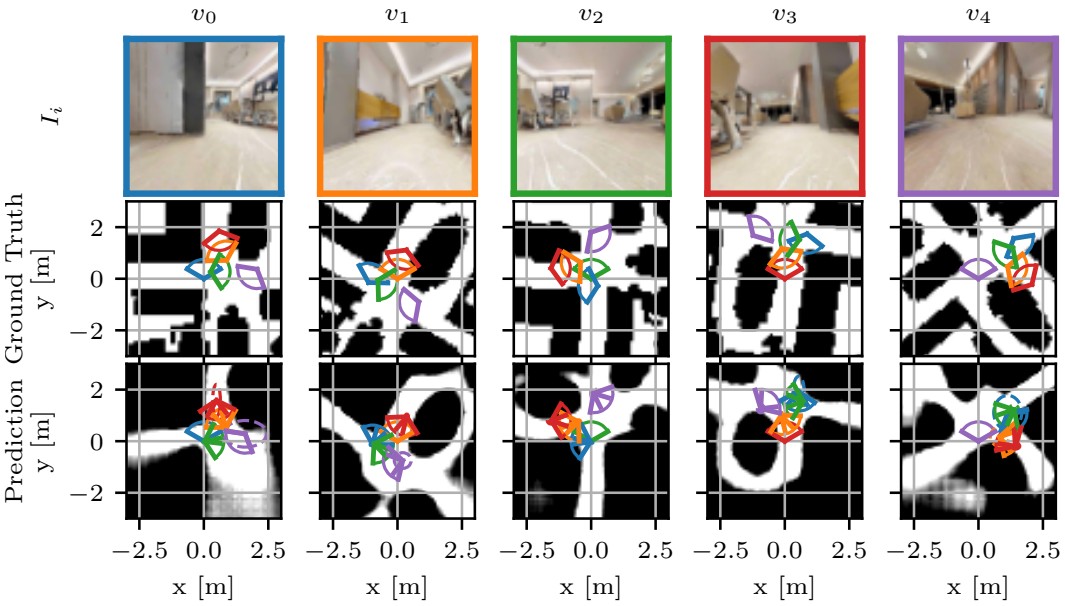

Figure 13: Sample D from $\mathcal{D}_{\text{Test}}^{\text{Sim}}$.

**6D Dataset:** We train an additional set of seven models with differing $S$ and $F$ on the dataset $\mathcal{D}_{\text{Train6D}}^{\text{Sim}}$. Notably, this dataset contains fully randomized 6D camera poses (opposed to roll and pitch constrained to $\pi/32$ in $\mathcal{D}_{\text{Train}}^{\text{Sim}}$). We furthermore randomize the camera's FOV. We show the results in Tab. 4. While the simulated results are not comparable to the one reported in Tab. 1 as the dataset is different, the real-world dataset is identical, and the reported performance is approximately identical with a slight degradation. The rotation error on $\mathcal{D}_{\text{Test6D}}^{\text{Sim}}$ is higher as there is a larger number of cameras without any direct visual overlap, and the maximum possible pose error increases.

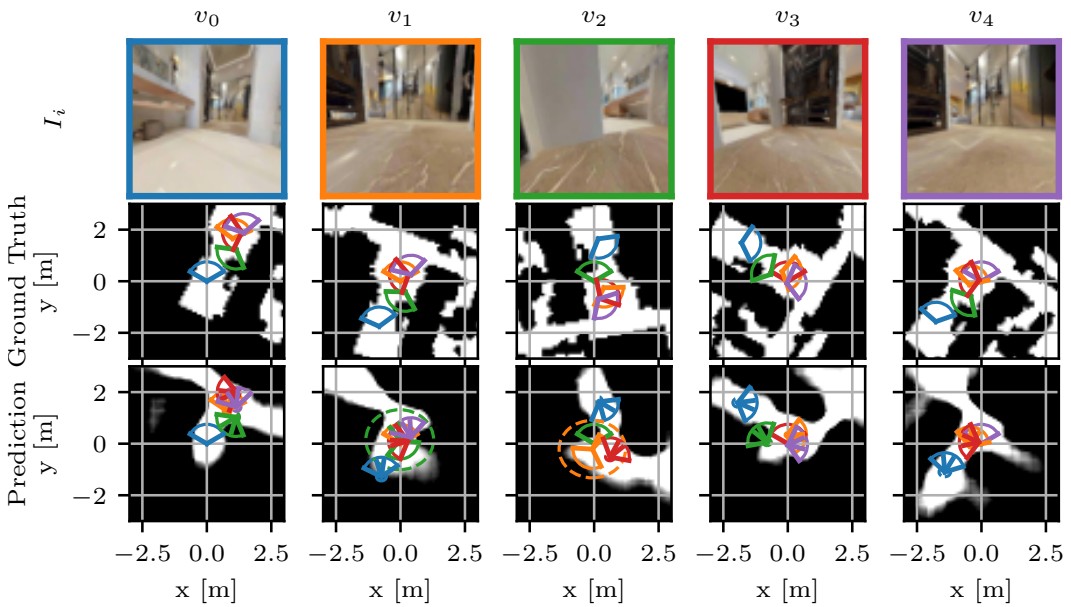

Figure 14: Sample E from $\mathcal{D}_{\text{Test}}^{\text{Sim}}$.

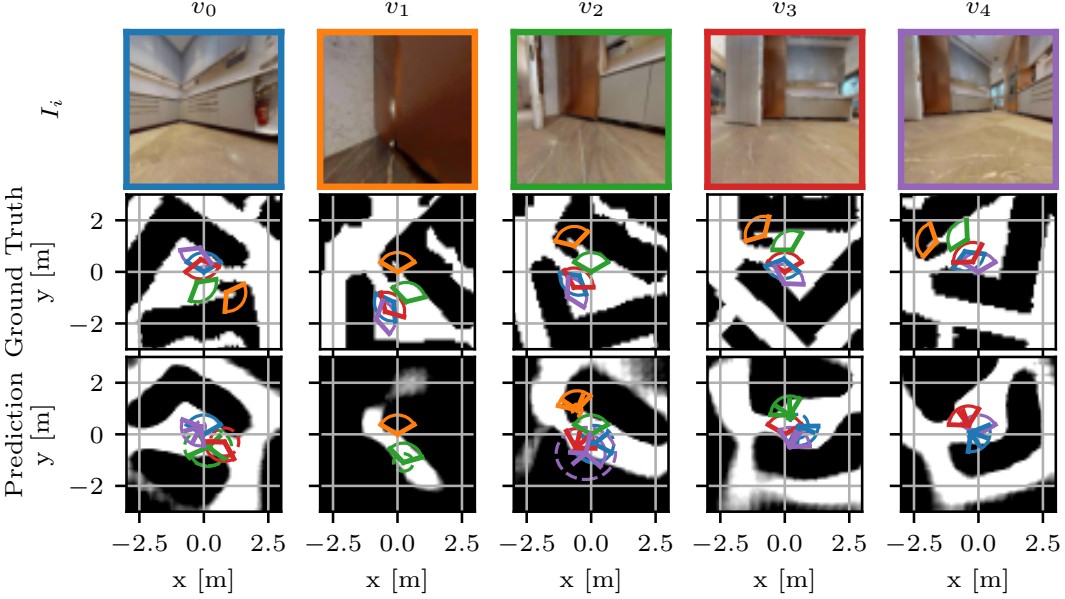

Figure 15: Sample F from $\mathcal{D}_{\text{Test}}^{\text{Sim}}$.

## B.3 Real-World

We provide additional quantitative experiments in Fig. 16, Tab. 5 and Tab. 6. We visualize four additional qualitative samples from the real-world dataset $\mathcal{D}_{\text{Test}}^{\text{Real}}$ in Fig. 17 and Fig. 18.

We investigate the efficiency of the uncertainty estimation as well as the distributions of pose errors more closely. Fig. 16 illustrates the distributions of position and rotation errors, along with the predicted variances over the angle difference between robot pairs in the dataset $\mathcal{D}_{\text{Test}}^{\text{Sim}}$. An angle difference of $0°$ signifies robots facing the same direction, potentially resulting in significant image overlap, while an angle difference of $180°$ indicates opposite directions, with no overlap. The Field Of View (FOV) overlap threshold is highlighted in red, delineating errors with overlap (below the

Table 4: Ablation study over the number of patches $S$ and size of features $F$ per patch for models trained on the dataset $\mathcal{D}_{\text{Train6D}}^{\text{Sim}}$. We report the BEV representation performance and the median error for poses on the dataset $\mathcal{D}_{\text{Test6D}}^{\text{Sim}}$ and $\mathcal{D}_{\text{Test}}^{\text{Real}}$.

| Model | | | | $\mathcal{D}_{\text{Test6D}}^{\text{Sim}}$ | | | | $\mathcal{D}_{\text{Test}}^{\text{Real}}$ | | | |
| S | F | Dice | IoU | All | | Invis. Filt. | | Invisible | | Visible | |
|---|---|---|---|---|---|---|---|---|---|---|---|
| 256 | 48 | 65.1 | 53.8 | **39 cm** | **46.5°** | **75 cm** | **9.3°** | **112 cm** | 16.0° | 44 cm | 7.1° |
| 128 | 96 | 64.8 | 53.3 | 42 cm | 46.6° | 100 cm | 13.3° | 119 cm | 16.8° | 41 cm | 6.8° |
| 128 | 48 | 64.9 | 53.4 | 43 cm | 46.6° | 88 cm | 13.0° | 113 cm | **14.9°** | **41 cm** | **6.5°** |
| 128 | 24 | **66.8** | **54.1** | 48 cm | 47.2° | 107 cm | 23.7° | 120 cm | 23.3° | 45 cm | 7.0° |
| 64 | 48 | 64.3 | 52.7 | 52 cm | 47.9° | 138 cm | 16.0° | 126 cm | 16.2° | 48 cm | 7.1° |
| 32 | 24 | 66.0 | 53.5 | 59 cm | 48.8° | 166 cm | 15.4° | 127 cm | 19.8° | 55 cm | 7.5° |
| 1 | 48 | 58.3 | 46.4 | 106 cm | 58.5° | 153 cm | 101.6° | 132 cm | 66.4° | 102 cm | 24.3° |

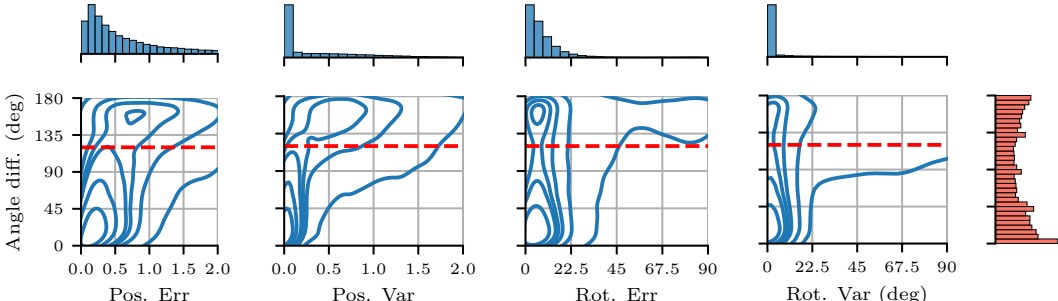

Figure 16: We show qualitative distributions for position and rotation errors and variances over the relative angle between two robots on all edges of the real-world testset $\mathcal{D}_{\text{Test}}^{\text{Real}}$, containing 84048 unique edges. Two agents facing the same direction would have an angle difference of $0°$, whereas two agents facing the opposite direction would have an angle difference of $180°$. We show the FOV as dashed line, where any results below the line indicate that the edge has some image overlap and above no image overlap. The horizontal marginals show the distribution of each individual plot while the vertical marginal shows the distribution of angle differences over the testset. Note that the angle distribution is not uniform due to the real-world dataset collection where at least two cameras always face the opposite direction.

line) from those without (above the line). The analysis reveals that position errors remain consistent up to the FOV threshold, beyond which they escalate, as reflected in the predicted position variance. Conversely, rotation errors stay uniform across angle differences, with only a marginal increase for higher angles, though the predicted rotation variance notably rises.

In Tab. 5, we investigate the performance of our model across the different scenes in the testing dataset $\mathcal{D}_{\text{Test}}^{\text{Real}}$. We provide a detailed breakdown of the median pose error for Invisible, Invisible Filtered and Visible cases for each of the eight scenes. While the pose error for visible samples is consistently between 22 cm/5.2°and 48 cm/7.5°, there are more outliers for invisible samples. This is due to some scenes having more or less symmetry, thus making it more challenging for the model to accurately predict poses. The pose prediction for the outdoor scene are least accurate, which is to be expected since the model was trained on indoor scenes. The Sunny scene has severe shadows and direct sunlight, while still providing low pose errors. The pose predictions are useful in all scenarios, demonstrating impressive generalization ability across a wide range of scenes.

Next, we evaluate the magnitude of pose prediction errors over different thresholds of distances between image pairs in Tab. 6. We consider five bands of distances separated in 0.4 m chunks, ranging up to 2m. Note that the worst-case pose error is $2 \cdot d$ due to the three-dimensional pose predictions. While the error increases with an increasing distance between samples, the rotational error in particular stays low, and the position errors stay useful, ranging from 12 cm to 65 cm for visible samples, 34 cm to 134 cm for invisible filtered samples (samples that are not visible but for which he model predicts a high confidence) and 76 cm to 153 cm for all invisible samples.

Table 5: The median pose estimation error of our model on each of the eight $\mathcal{D}_{\text{Test}}^{\text{Real}}$ scenes.

| Scenario | Invisible | | Invisible Filtered | | Visible | |
|---|---|---|---|---|---|---|
| Corridor A | 138 cm | 129.6° | 90 cm | 11.4° | 22 cm | 5.2° |
| Corridor B | 112 cm | **5.9°** | 261 cm | 8.2° | 31 cm | 5.2° |
| Office A | 86 cm | 14.4° | **38 cm** | 9.6° | 33 cm | 7.5° |
| Office B | 146 cm | 10.4° | 248 cm | **4.6°** | **20 cm** | **4.7°** |
| Outdoor | 134 cm | 9.5° | 193 cm | 28.7° | 49 cm | 6.9° |
| Study A | 132 cm | 6.7° | 40 cm | 5.1° | 28 cm | 5.3° |
| Study B | 115 cm | 8.9° | 56 cm | 9.6° | 34 cm | 5.5° |
| Sunny | **77 cm** | 9.3° | 49 cm | 8.7° | 22 cm | 6.0° |

Table 6: We show median pose prediction errors for different distance thresholds $d$ on the dataset $\mathcal{D}_{\text{Test}}^{\text{Real}}$.

| $d$ | Invisible | | Invisible Filtered | | Visible | |
|---|---|---|---|---|---|---|
| <0.4 m | **76 cm** | **7.6°** | **34 cm** | **6.7°** | **12 cm** | **4.9°** |
| <0.8 m | 80 cm | 8.8° | 48 cm | 8.6° | 21 cm | 5.2° |
| <1.2 m | 95 cm | 9.5° | 56 cm | 8.5° | 31 cm | 5.8° |
| <1.6 m | 123 cm | 10.6° | 97 cm | 12.9° | 45 cm | 6.0° |
| <2.0 m | 153 cm | 10.5° | 134 cm | 16.2° | 65 cm | 6.8° |

## B.4   Real-World Pose Control

**Ad-hoc WiFi**: Image embeddings are sent between nodes at a rate of 15hz, with each being just over 6KiB. The fully distributed nature of our system matches well with a broadcast communications topology, which theoretically can permit operation at scale. Broadcast messaging has a significant drawback however, in that modern wireless data networking standards (IEEE 802.11/WiFi) fallback to very low bit rates to ensure maximum reception probability.

802.11 uses low bit rates for broadcast due to a lack of acknowledgment messages, which are the typical feedback mechanism for data rate control and retransmissions. The probability of a frame being lost due to the underlying CSMA/CA frame scheduler is non-trivial (3%+); a condition which worsens with increasing participating network nodes.

A higher level system, such as ROS2 communications middleware package CycloneDDS can manage retransmissions, however currently available systems are not well suited to the requirements of a decentralized robotic network. In particular they lack the ability to detect network overload conditions, which can result in retransmissions that further load the network; finally resulting in a rapid increase in packet loss rates which renders the network as a whole unusable. While it is possible to tune such communications packages manually, the best parameter values are sensitive to highly variable deployment configuration parameters including ambient wireless network traffic. Manual tuning of network hardware, such as forcing increased broadcast bit rates, can alleviate this problem but are generally not resilient as scale.

Our custom messaging protocol is targeted to this specific use case, and includes the ability to dynamically backoff messaging load based upon detected network conditions. Our system is also uses a shared slot TDMA message scheduling system which works with the 802.11 frame scheduler to maximize frame reception probability without compromising overall data rates.

**Results**: We benchmark the runtime performance of the model on the Jetson Orin NX in Tab. 7 by averaging 100 sequential forward passes. We compare 16-bit and 32-bit float evaluated on the GPU or CPU and compilation with TensorRT, a proprietary NVidia tool for ahead-of-time compilation of neural networks. The model $f_{\text{enc}}$, generating the communicated embedding $\mathbf{E}_i$, has a total of 30 M parameters, and takes ca. 20 ms for one forward pass as FP16 optimized with Torch TensorRT on the GPU. This is $50\times$ faster than running the same model on the CPU. We also note that changing from 32-bit floats to 16-bit floats results in a $2.75\times$ speedup. The model $f_{\text{pose}}$, processing a pair of

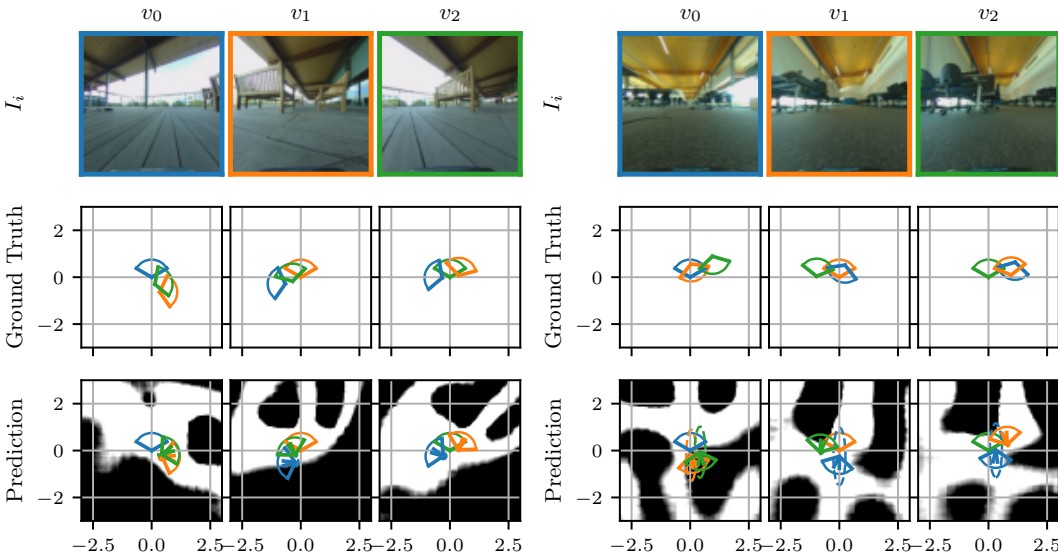

Figure 17: We visualize a sample from our custom real-world testset $\mathcal{D}_{\text{Test}}^{\text{Real}}$ for three nodes, similar to the simulation sample in Fig. 2. We do not show the ground-truth BEV segmentation since we do not have labels for this. Additionally to the pose predictions $\hat{\mathbf{p}}_{i,ij}, \hat{\mathbf{R}}_{i,ij}$, we also visualize the predicted uncertainties $\sigma_{\text{p},i}^2 ij$ and $\sigma_{\text{R},i}^2 ij$.

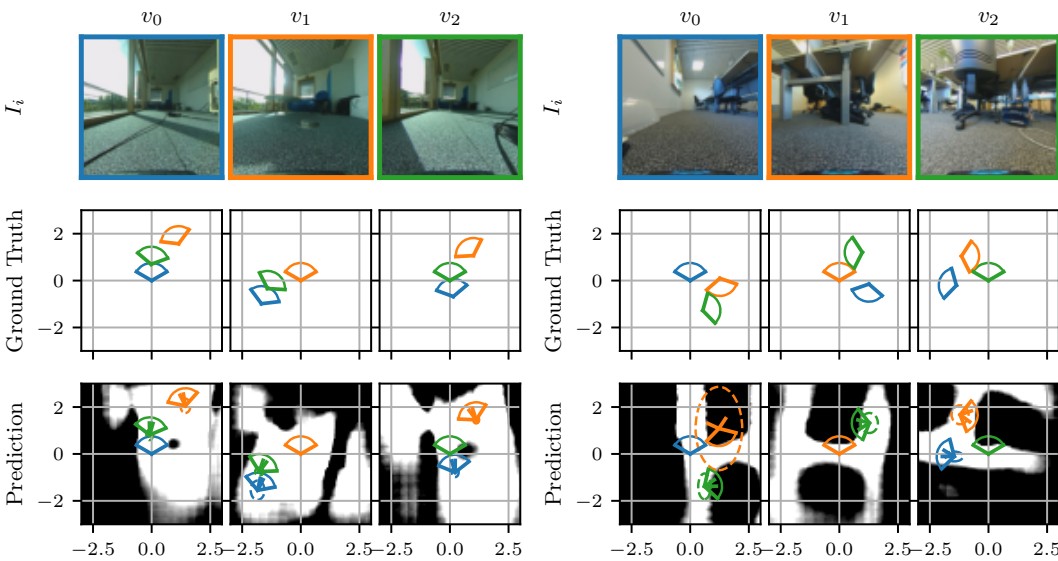

Figure 18: Sample B and C from $\mathcal{D}_{\text{Test}}^{\text{Real}}$.

embeddings into a relative pose, has a total of 6 M parameters and takes ca. 4 ms for one forward pass. The model is run in a custom C++ environment integrated with ROS2 Humble and an ad-hoc WiFi network for inter-node communication to communicate image embeddings $\mathbf{E}_i$, achieving a 15 Hz processing rate for the whole pipeline, including image pre-processing, model evaluation, communication and pose prediction.

Fig. 20 and Tab. 8 contain additional quantitative and qualitative evaluation for the multi-robot pose-tracking experiment, consisting of two additional reference trajectories. Figure 8 static is similar to the Figure 8 dynamic trajectory elaborated in the main paper, but all robots face the same direction, resulting in two shifted Figure 8s for the two follower robots. Rectangle dynamic is also similar to

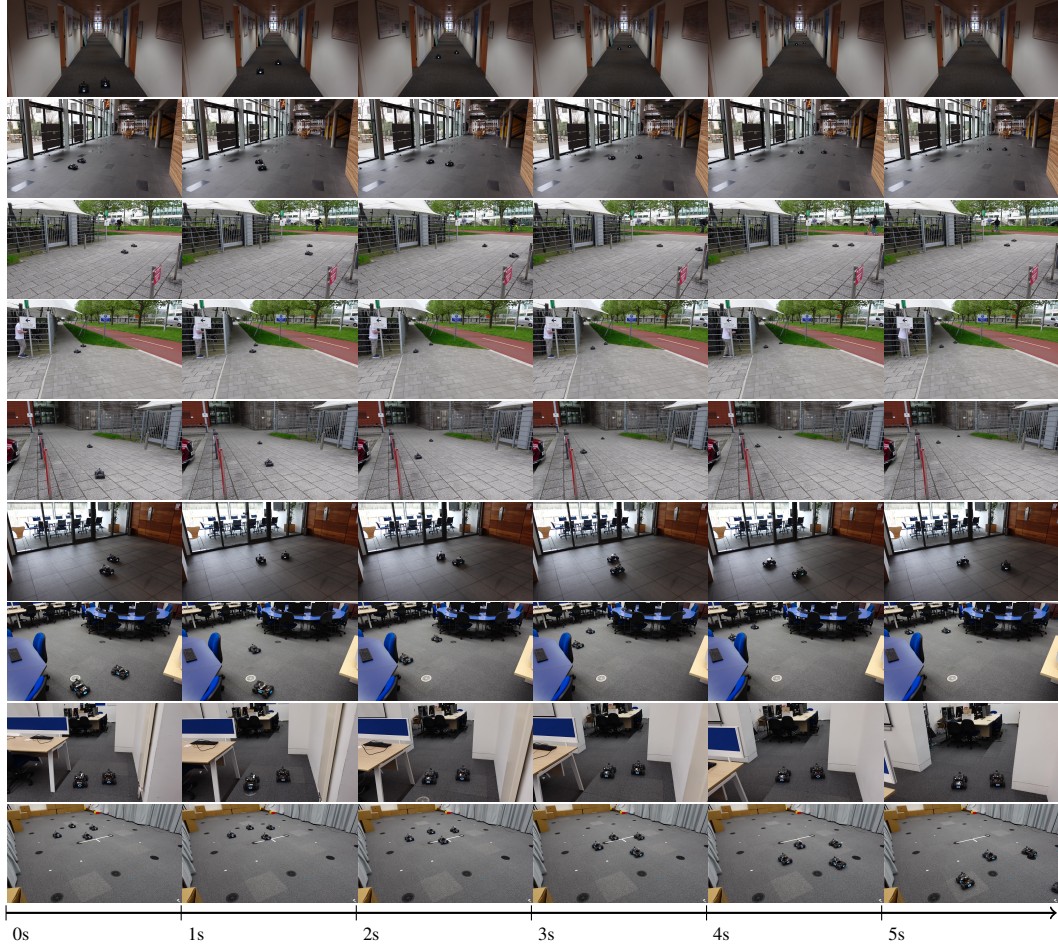

Figure 19: We show additional snapshots of real-world deployments in different scenes with up to four robots. Each row contains six frames from a video recording, each spaced $1\,\text{s}$ in time.

Table 7: Model runtime evaluation for $S = 128$ and $F = 24$.

| Type | Mode | Dev. | $f_{\text{enc}}$ | | $f_{\text{pose}}$ | |
|------|------|------|------|------|------|------|
| FP16 | TRT | GPU | **19.22 ms** | $\pm 0.92$ ms | **4.15 ms** | $\pm 0.49$ ms |
| FP16 | JIT | GPU | 36.79 ms | $\pm 1.33$ ms | 8.04 ms | $\pm 0.73$ ms |
| FP32 | JIT | GPU | 99.32 ms | $\pm 1.56$ ms | 13.57 ms | $\pm 1.10$ ms |
| FP32 | JIT | CPU | 952.04 ms | $\pm 116.72$ ms | 248.41 ms | $\pm 45.98$ ms |

the Figure 8 dynamic trajectory elaborated in the main paper, but instead of a Figure 8, the reference is a rounded rectangle, with robots moving with up to 0.8 m/s.

Tab. 8 reports quantitative evaluations for the same results, which are in line with the results reported in Tab. 1. Note that the velocities reported are average velocities of the follower robots. The velocities are consistent for both Figure 8 trajectories, but differ for the rectangle trajectory, as the inner robot moves more slowly than the outer robot.

## B.5 Single-Robot Trajectory Recording

We perform a single-robot homing/trajectory replay experiment to further show the versatility of CoViS-Net. In this experiment, we treat a list of recorded keyframes as reference embedding to estimate the relative pose to given the currently observed image. If either the distance or the predicted

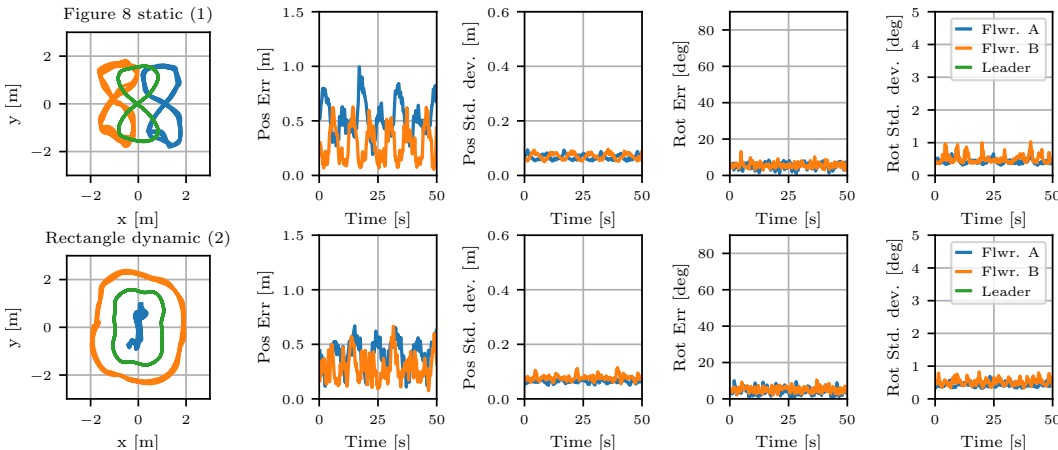

Figure 20: We evaluate the tracking performance of our model and uncertainty-aware controller on two additional reference trajectories, with two follower robots (in blue and orange), positioned left and right of the leader robot (in green). For each trajectory, we report the tracking performance over time for position and rotation, as well as the predicted uncertainties. The top trajectory (1) is a Figure Eight with the leader always facing in the same direction. The bottom trajectory (2) is a rounded rectangle. We show the trajectory for 120 s and the tracking error for the first 60 s. The tracking performance is consistent across multiple runs.

Table 8: We report the mean and absolute tracking error for both leaders for all three trajectories, as well as average velocities. The errors are consistent with the results reported in Tab. 1.

| Trajectory | Robot | Mean Abs. | Median | Vel |
|---|---|---|---|---|
| Figure 8 dynamic | A | 38 cm, 5.6° | 38 cm, 4.8° | 0.59 m/s |
|  | B | 28 cm, 5.1° | 25 cm, 4.7° | 0.58 m/s |
| Figure 8 static | A | 51 cm, 5.2° | 48 cm, 5.3° | 0.60 m/s |
|  | B | 28 cm, 5.6° | 26 cm, 5.4° | 0.61 m/s |
| Rectangle dynanic | A | 41 cm, 4.4° | 42 cm, 4.4° | 0.32 m/s |
|  | B | 29 cm, 5.1° | 27 cm, 5.1° | 0.81 m/s |

uncertainty increases by some predetermined threshold, we append a new keyframe to the list. After teaching the robot the reference trajectory, we command it into a keyframe-following mode, where the current reference keyframe is set based on the recorded list. We provide qualitative experiment on the website.

