# OpenReview forum: "CoViS-Net: A Cooperative Visual Spatial Foundation Model for Multi-Robot Applications"
_robot-learning.org/CoRL/2024/Conference — CoRL 2024_

### Official Review · Reviewer_mxQi · 2024-07-17
**A decentralized, platform-agnostic visual spatial model for multi-robot system.**

**Originality:** 3
**Technical Quality:** 3
**Clarity Of Presentation:** 3
**Potential Impact:** 3
**Recommendation:** 3
**Confidence:** 4

**Review:**

[Strengths]
1. Innovative Approach: The use of a transformer-based architecture for pose prediction is novel and interesting. The integration of visual and inertial data through a cooperative network enhances pose estimation accuracy, as evidenced by the experimental results.

2. Comprehensive Evaluation: The authors provide a thorough evaluation of their model, including both simulated and real-world datasets. The use of different metrics (Euclidean distance, geodesic distance) to report errors adds credibility to their findings.

3. Baseline Comparisons: The inclusion of baseline comparisons with ORB and LightGlue methods effectively highlights the superior performance of CovisNet. This comparative analysis is crucial for establishing the novelty and effectiveness of the proposed method.

[Weaknesses]

The characterization of CoViS-Net as a "foundation model" warrants a closer examination. Here are several points that challenge the classification of CoViS-Net as a true foundation model.

1. Scope of Training Data: Foundation models are typically trained on extensive and diverse datasets that cover a wide array of scenarios and contexts. While CoViS-Net uses a substantial dataset, including the HM3D dataset and simulations, its focus remains narrow within the domain of multi-robot systems. The training data is tailored to specific robotic tasks, limiting its generalizability.

2. Specialized Architecture: The architecture of CoViS-Net, while innovative, is specialized for the task of relative pose estimation and BEV representation in multi-robot systems. Foundation models usually feature a more generic architecture that can be adapted for multiple, diverse tasks. The specialized nature of CoViS-Net’s architecture suggests a more application-specific model rather than a general-purpose foundation model.

3. Scale: Foundation models are often exceptionally large, with billions of parameters, enabling them to capture vast amounts of knowledge and nuances from their training data. CoViS-Net, while complex, does not reach the scale typically associated with foundation models. Its parameter count and computational requirements are more in line with specialized deep learning models rather than the large-scale models seen in NLP or vision foundation models.

**Quality Of The Limitations Section:**

3

**Questions For Rebuttal:**

Could the authors provide a clear definition of what they consider a "foundation model" and how CoViS-Net fits into this category?

How does CoViS-Net compare to well-known foundation models in terms of architecture, training scope, and adaptability?

What is the extent of diversity in the training data used for CoViS-Net? How do the authors ensure that the model generalizes well beyond the specific scenarios presented in the paper?

What is the parameter count of CoViS-Net, and how does it compare to traditional foundation models? Can the authors justify the scale of the model in the context of foundation models?

**Robotics Focus:**

4

**Summary Of Paper:**

This paper introduces a transformer-based architecture to enable accurate real-time pose estimation and spatial understanding in multi-robot systems, validated through comprehensive simulations and real-world experiments.

**Summary Of Recommendation:**

The specialized nature, limited training scope, and lack of demonstrated versatility across tasks suggest that CoViS-Net is better classified as a highly specialized model rather than a general-purpose foundation model. This distinction is important for setting appropriate expectations and understanding the potential and limitations of the proposed approach.

---

### Official Review · Reviewer_bCfq · 2024-07-19

**Originality:** 3
**Technical Quality:** 3
**Clarity Of Presentation:** 4
**Potential Impact:** 3
**Recommendation:** 4
**Confidence:** 5

**Review:**

## Quality
The paper presents several strong technical contributions. The ablation study effectively motivates key components of the model, providing insights into the importance of each aspect. The inclusion of real-world experiments to validate the model’s performance significantly enhances the credibility of the proposed approach, demonstrating its practical applicability. However, the paper could benefit from a scalability analysis, particularly in larger multi-robot systems, and the potential challenges and limitations associated with it. The benchmarking section, which compares CoViS-Net with only two feature matching methods, is somewhat limited. Adding a third benchmark will help to better highlight the advantages of the proposed model.

## Clarity
The paper is generally well-structured and easy to follow, with a clear flow of information from the introduction to the experimental results. However, the literature review could be improved by discussing related work on map prediction methods and how BEV representations relate to these similar approaches. Specifically, citing necessary map prediction papers such as:
- 4CNet: A Confidence-Aware, Contrastive, Conditional, Consistency Model for Robot Map Prediction in Multi-Robot Environments, 2024
- Enhancing robot task completion through environment andtask inference: a survey from the mobile robot perspective, 2022
- Distributed Inference based multi-robot exploration, 2018

Additionally, other applications in the literature review should be included, such as multi-robot exploration/navigation/search:
- NavFormer: A Transformer Architecture for Robot Target-Driven Navigation in Unknown and Dynamic Environments, 2024
- Deep Reinforcement Learning for Decentralized Multi-Robot Exploration with Macro Actions,2023
- A multirobot person search system for finding multiple dynamic users in human-centered environments, 2023

A more detailed explanation of why feature matching benchmarks were chosen for comparison would improve clarity. While the paper proposes a method for improving relative pose estimations and BEV representations, the comparison with feature matching methods like ORB and LightGlue is not entirely clear. This table should also include FPS to see how the proposed method compares in terms of performance. It is also not clear why the ablation and comparison methods use different metrics, which could be. As Euclidean distance between the predicted and ground truth poses and the geodesic distance between rotations seem like more direct metrics when evaluating pose estimation and BEV representation.

## Originality
Using BEV representations to improve relative pose estimation is relatively original. However, the paper does not sufficiently compare this approach to other map prediction methods or tasks such as robot exploration, which could potentially address the same issues. A more detailed comparison with these areas would better highlight the novelty and advantages of CoViS-Net.

## Significance
The paper is significant because it provides a robot/platform-agnostic solution to multi-robot pose estimation and spatial understanding without requiring external networking infrastructure. Its real-time processing capabilities make it highly practical for real-world applications.

## Strengths:
- Effective ablation study to motivate key components.
- Strong real-world validation with detailed experimental analysis.
- Incorporation of uncertainty estimation.
- Robot/platform-agnostic approach that does not require external networking infrastructure.
- Real-time processing capabilities.

## Weaknesses:
- Improve the literature review by including additional multi-robot search and exploration papers, focusing on map prediction methods and adding a scalability analysis.
- Ensure comparison metrics are consistent or clearly justified, and include an ablation study comparing with and without BEV generation to highlight its benefits.

**Quality Of The Limitations Section:**

1

**Questions For Rebuttal:**

- The literature review needs improvement. Can the authors discuss related work on map prediction methods and how BEV representations relate to these approaches? Additionally, include relevant studies on multi-robot exploration and search.
- The use of different metrics for ablation and comparison methods is confusing. Can the authors streamline the metrics and use consistent measures such as Euclidean distance and geodesic distance for evaluating pose estimation and BEV representation?\
- Can the authors include an ablation study comparing results with and without BEV generation to highlight its benefits?
- Please include limitations of your approach.

**Robotics Focus:**

4

**Summary Of Paper:**

The paper proposes CoViS-Net, a decentralized, platform-agnostic visual spatial foundation model designed for multi-robot applications in unstructured environments. CoViS-Net leverages spatial priors learned from data to provide real-time pose estimation and bird’s-eye-view (BEV) representations, even without camera overlap between robots. The model operates on onboard compute without requiring existing networking infrastructure.

**Summary Of Recommendation:**

This work demonstrates several strengths, including an effective ablation study that underscores the importance of key components, strong real-world validation with comprehensive experimental analysis, and the incorporation of uncertainty estimation. The approach is robot/platform-agnostic, does not rely on external networking infrastructure, and supports real-time processing capabilities.     However, there are some areas for improvement. The literature review should be enhanced by including more papers on multi-robot search and exploration, with a particular focus on map prediction methods and scalability analysis. Additionally, comparison metrics should be made more consistent or clearly justified, and an ablation study should be conducted comparing scenarios with and without BEV generation to highlight its benefits.  I recommend weak accept.

---

### Official Review · Reviewer_vrDg · 2024-07-23
**Review of Submission 176**

**Originality:** 2
**Technical Quality:** 3
**Clarity Of Presentation:** 2
**Potential Impact:** 2
**Recommendation:** 3
**Confidence:** 2

**Review:**

Overall, I think the proposed method is interesting, and there could be some technical insights in the methodology (as far as I can tell, it seems sound). The paper includes a nice demonstration on real hardware and lots of technical details (probably more than necessary for a CoRL paper).

Strengths:
- The idea to use DINO features from multiple robots as an intermediate encoding representation for multi-robot relative pose estimation seems reasonable.
- The model design and loss seem reasonable to me,
- The method is tested on real hardware, which is great to see.

Weaknesses:
- There are many seemingly misleading claims in the paper, including the presentation of this work as a "foundation model." As I understand it, this work uses pre-trained DINO features as an encoding. DINO itself is a "foundation model" in the common usage, but I'm not sure that this makes every downstream application which uses DINO immediately a "foundation model" as well. Calling the proposed model a foundation model would seem to imply that the proposed model itself is trained on a broad, large-scale dataset. As far as I can tell, this is not the case, but maybe I am missing something.
- Similarly, in the abstract the paper states "classical vision-based methods to regress relative pose are commonly computationally expensive (precluding real-time applications)." I suspect I must have misunderstood this claim, because I believe multi-robot SLAM broadly (and the related technologies it encompasses) is a relatively well understood area of research at this point. This includes, for example, visual feature matching / place recognition and relative pose estimation from feature correspondences. These problems are by no means _solved_, but computational expense does not seem to me to be the primary issue precluding the widespread use of these systems.
- The presentation seems unnecessarily verbose. It is clear that the authors have a lot of confidence in the proposed approach, and they want to communicate their enthusiasm, but occasionally I felt that too much information was being thrown out at once, and it would serve the paper well to more concisely highlight just those key ideas which are most relevant to the proposed work. To give some specific examples to guide potential revisions, phrases like: "a decentralized, platform-agnostic visual spatial foundation model designed for real-world multi-robot applications," "a deep distributed stateless architecture to tackle multi-robot uncertainty-aware relative pose estimation," "platform-agnostic, decentralized, real-time multi-robot pose estimation from monocular images," really make it hard as a reader to identify what the most salient aspect(s) of the proposed contribution is.
- One of the major proposed benefits of the presented work is that it will attempt to estimate a relative pose even between non-overlapping images based on an imagined context. While this is demonstrated to work within the scope of the presented experimental results, it's not actually obvious to me that this is a desirable property of such a system. I still think the method is interesting and there could be some benefits to predicting local context outside the field of view (for example for local planning), but it seems for practical purposes that it might be better in those instances to simply recognize that there is no overlap and not attempt to register the relative pose between the robots.
- The "Limitations and Conclusions" section does not appear to contain any description of the limitations of the proposed system.

**Quality Of The Limitations Section:**

1

**Questions For Rebuttal:**

See comments above

**Robotics Focus:**

4

**Summary Of Paper:**

This paper presents a method for estimating relative poses between robots and "birds eye view" map representations in the setting of multi-robot navigation. The model is based on a neural network that uses pre-trained foundation model features (DINO) to predict relative poses and birds-eye view representations. The model is tested in real multi-robot experiments.

**Summary Of Recommendation:**

I am recommending "weak accept" on the basis that the method itself seems interesting and the technical evaluation seems reasonable to me, although some aspects of the presentation could be clarified.

---

### Official Review · Reviewer_3vko · 2024-07-24
**A paper with low theoretical novelty but good efforts to make the approach run in real-time**

**Originality:** 2
**Technical Quality:** 3
**Clarity Of Presentation:** 3
**Potential Impact:** 3
**Recommendation:** 2
**Confidence:** 5

**Review:**

Strengths:

+ Efforts to make the approach run in real-time on a robot's onboard GPU are appreciated.

+ The video demonstrations on real robots are illustrative.

+ Overall, the paper is easy to follow.

Weaknesses:
- The theoretical novelty of this paper is low. The main contribution to relative pose estimation in non-overlapping cases relies heavily on the pre-trained DinoV2 model. The loss function for uncertainty quantification is also widely used.

- It is not clear how the proposed method addresses non-overlapping multi-robot views for relative pose estimation. If two images have no overlap, why does the method work and can maintain small rotation uncertainty. Line 190 states, "...allows it to 'imagine' unseen portions of the scene..." What theoretical and experimental supports justify this statement?

- Visual cameras alone do not provide direct measurements of the absolute scale of objects or the distance between them. How does the proposed method address scale issues?

- How can the quantified uncertainties be used to improve pose estimation, especially given that DinoV2 is a component?

- How is the proposed method decentralized? Is it achieved by running the same full model on each robot?

- It appears that the experiments evaluate pairwise situations where each follower follows the leader. How can the proposed approach generalize to estimating poses of more than one robot? How does it handle potential conflicting pose estimation results (e.g., when two robots try to estimate each other's poses)?

- Minor: Why is the proposed method referred to as a foundation model? Using a pre-trained DinoV2 as an image encoder does not make the proposed method a foundation model. This terminology seems overstated and misleading.

**Quality Of The Limitations Section:**

1

**Questions For Rebuttal:**

Please see the review section.

**Robotics Focus:**

4

**Summary Of Paper:**

The paper introduces a method for relative pose estimation with uncertainty quantification using monocular visual images. The method is built upon DinoV2 for image encoding, where DinoV2 can address relative pose estimation without overlapping views. A pairwise pose encoder is then used to estimate the relative poses, and multi-node aggregation is utilized to construct a BEV representation. In addition, the uncertainty in the relative pose estimation is quantified using the Gaussian NLL loss. The lightweight model design and GPU acceleration allow the method to run onboard in real-time.

**Summary Of Recommendation:**

The theoretical novelty of the paper is limited, and several key aspects of the approach, such as addressing non-overlapping views and scale issues, are not well explained or justified. The efforts to achieve real-time performance are commendable.

---

### Author Rebuttal · Authors · 2024-08-12

We thank all reviewers for their time and effort in reviewing our paper and providing valuable feedback. We have addressed comments on conciseness and verbosity, highlighted our contribution, and added suggested papers to our literature review. We ran additional experiments to evaluate the speed of our baselines and added these results to Table 2. Lastly, we added a limitations section. For the reviewer's convenience, all changes are highlighted.

---

### Decision · Program_Chairs · 2024-09-04

**Decision:**

Accept

**Comment:**

Strength:

- Reviewers praised the paper's focus on real-time performance and the demonstration on real robots. (Reviewer 3vko, Reviewer vrDg, Reviewer bCfq)
- Comprehensive evaluation: The paper includes both simulation and real-world experiments, uses multiple metrics, and compares against baseline methods, strengthening the results' validity. (Reviewer mxQi)
- Novel use of BEV representations: Employing BEV representations for improving relative pose estimation was deemed relatively original. (Reviewer bCfq)

Weaknesses:

- Limited theoretical novelty: Several reviewers found the paper's reliance on existing techniques
- Unclear justification for “foundation model" claim: The paper's characterization of its model as a "foundation model" drew criticism. Reviewers questioned the training data scope, specialized architecture, and lack of comparison with established foundation models. (Reviewer vrDg, Reviewer mxQi)

After rebuttal, the majority of the reviews lean towards accepting this paper. This work has strong real world evaluation and demonstration, while needs more polishing in presentation and positioning. I would recommend accepting this paper and encourage the authors to further improve the paper according to the suggestions.